

# Cosmogenic ³He paleothermometry on post-LGM glacial bedrock within the central European Alps

Natacha Gribenski[1,2], Marissa M. Tremblay[3], Pierre G. Valla[4], Greg Balco[5], Benny Guralnik[6], David L. Shuster[5,7]

[1]Institute of Geological Sciences, University of Bern, Bern, 3012, Switzerland
[2]Oeschger Centre for Climate Change Research, University of Bern, Bern, 3012, Switzerland
[3]Department of Earth, Atmospheric, and Planetary Sciences, Purdue University, West Lafayette, IN 47901, USA
[4]University Grenoble Alpes, University Savoie Mont Blanc, CNRS, IRD, IFSTTAR, ISTerre, Grenoble, 38000, France
[5]Berkeley Geochronology Center, Berkeley, CA 94709, USA
[6]Technical University of Denmark, Kgs. Lyngby, 2800, Denmark
[7]Department of Earth and Planetary Science, University of California, Berkeley, CA 94709, USA

*Correspondence to*: Natacha Gribenski (natacha.gribenski@geo.unibe.ch)

**Abstract.** Diffusion properties of cosmogenic ³He in quartz at Earth's surface temperatures offer the potential to reconstruct the evolution of past in-situ temperatures directly from formerly glaciated areas, information important for improving our understanding of glacier-climate interactions. In this study, we apply cosmogenic ³He paleothermometry on rock surfaces gradually exposed since the Last Glacial Maximum (LGM) to the Holocene period along two deglaciation profiles in the European Alps (Mont Blanc and Aar massifs). Laboratory experiments conducted on one representative sample per site indicate significant variability in ³He diffusion kinetics between the two sites, with quasi linear Arrhenius behavior observed in quartz from the Mont Blanc site and complex Arrhenius behavior observed from the Aar site, which we interpret to indicate the presence of multiple diffusion domains (MDD). Assuming that same diffusion kinetics apply to all quartz samples along each profile, predictive simulations indicate that ³He abundance in all the investigated samples should be at equilibrium with present-day temperature conditions. However, measured natural ³He concentrations in samples exposed since before the Holocene indicate an apparent ³He thermal signal significantly colder than today. This observed ³He thermal signal cannot be explained with a realistic post-LGM mean annual temperature evolution in the European Alps at the study sites. One hypothesis is that the diffusion kinetics and MDD model applied may not provide sufficiently accurate, quantitative paleo-temperature estimates in these samples; thus, whereas pre-Holocene ³He thermal signal is indeed preserved in the quartz, the helium diffusivity would be lower at Alpine surface temperatures than our diffusion models predict. Alternatively, if the modeled helium diffusion kinetics is accurate, the observed ³He abundances may reflect complex geomorphic/paleoclimatic evolution with much more recent ground temperature changes associated with the degradation of alpine permafrost.





## 1 Introduction

This study applies cosmogenic noble gas paleothermometry (Tremblay et al., 2014) to attempt to reconstruct temperature changes associated with gradual ice lowering following the Last Glacial Maximum (LGM ca. 27-19 ka; Clark et al., 2009) in two sites of the high European Alps. Because glaciers are sensitive to both temperature and precipitation, obtaining information about *in situ* temperature conditions from an independent proxy is critical to disentangling the role of either variable in recorded

glaciers fluctuations and to adequately use these records for paleoclimate reconstructions. In particular, paleoglacier records can then be used as direct site-specific paleo-precipitation indicators (e.g., Kerschner et al., 2000; Kerschner and Ivy-Ochs, 2008; Martin et al., 2020) to trace changes in regional atmospheric circulation systems (Kuhlemann et al., 2008; Becker et al., 2016; Gribenski et al., 2021). More detailed information about paleoclimate conditions would moreover improve our understanding of glacier response(s) to current climate change as well as our ability to anticipate glacier evolutions for

proposed future climate scenarios (Zemp et al., 2006; Haeberli et al., 2020). Furthermore, direct temperature constraints associated with paleoglacier variations are also critical to our understanding of glacier erosion processes (Hallet, 1979) which have profoundly shaped high-latitude and mountain landscapes over $10^3$ to $10^6$ yr timescales (Herman et al., 2021), and which seems to relate, among other factors, to climatic conditions (Koppes et al., 2015; Cook et al., 2020).

Available data on the relationship between glacier geometry and climate as well as between glacial erosion and climate are

largely biased toward present-day and historical time periods, therefore obliging us to rely on the assumption that modern to centennial records are representative of the range of variation and mechanistic trends between climate/glacier variation and erosion operating on geological time scales (Jaeger and Koppes, 2016). While combined records of paleoglacier geometry and erosion rates on Late-Pleistocene timescales are growing thanks to the recent development of analytical and numerical techniques (e.g., Kapannusch et al., 2020; Mariotti et al., 2021), obtaining direct quantitative paleoclimate constraints from

formerly glaciated areas remains challenging, even for regions with relatively well known paleoglacial histories. In the European Alps, the most detailed paleoglacier record goes back to the Late Pleistocene ice maximum advance, dated around ~26-24 ka in the northern and central Alps (Monegato et al., 2017), in line with the global LGM. During the LGM, ice spread to within several tens of kilometers of the piedmonts and reached more than 1000-1500 m thick in the main valleys (Ivy-Ochs, 2015; Wirsig et al., 2016a). More restricted stages (i.e., Gschnitz, Daun, Egesen stadials; Ivy-Ochs, 2015) marking the gradual

retreat (and thinning) of the ice into the upper catchments followed between the LGM and the Younger Dryas cooling event (YD, 12.8-11.7 ka; Heiri et al., 2014a). During the early Holocene (i.e., the last 11 ka; Heiri et al., 2014a), glaciers retreated quickly behind the Little Ice Age moraines, and remained within these limits for the rest of the Holocene period (Heiri et al., 2014a).

The recorded Alpine glacial sequence is consistent with polar ice oxygen isotope ($\delta^{18}O$) records from the North hemisphere

(NGRIP, 2004), which indicate temperature minima reached at 25-20 ka (around a -20 °C anomaly in central Greenland compared to present mean annual temperatures), followed by a gradual increase until ca. 10 ka with the last pronounced isotopic excursion at ca. 12 ka (YD event, around -15°C anomaly; Buizert et al., 2018). After the YD, temperatures stabilized





around values similar to today with only minor fluctuations (less than 2°C) throughout the remaining Holocene period (Buizert et al., 2018). High-resolution Alpine $\delta^{18}$O speleothems similarly support a coupling between the northern European Alps and

Greenland records (Moseley et al., 2020; Li et al., 2021).

While there is evidence for temporal coupling, a direct (scaled) translation of polar ice records over the Alps to obtain quantitative temperature/precipitation constraints is inappropriate. Indeed, major climate forcing components (i.e., ice sheet, atmospheric greenhouse gas and ocean circulation) also underwent large-scale changes between the LGM and the Holocene transition (Clark et al., 2012), which resulted in a variable pre-Holocene latitudinal temperature gradient (Heiri et al., 2014b)

and North Atlantic atmospheric patterns (Eynaud et al., 2009). Existing past climate information from Alpine paleoenvironmental proxies is mainly qualitative with only a few scarce and fragmented quantitative temperature/precipitation records available for the pre-Holocene period (Heiri et al., 2014a). These are mostly from proxies located on the outer rim of the Alpine range from lake and peat archives (e.g., pollen, chironomids; Heiri et al., 2014a) and groundwater and speleothems (i.e., noble gas; Beyerle et al., 1998; Ghadiri et al., 2018) or tentatively-derived from inverse glacial modelling (Kerschner and

Ivy-Ochs, 2008; Becker et al., 2016; Seguinot et al., 2018), with some noticeable variability in derived paleoclimate information between and within proxies records. Proposed reconstructed mean temperature anomalies during the LGM hence vary from -11 to -14°C based on pollen reconstructions (Wu et al., 2007; Bartlein et al., 2014), -5 to -9°C based on noble-gas groundwater records (Beyerle et al., 1998, Seltzer et al., 2021), and -8 to -15 °C using glacial modelling studies calibrated on reconstructed ice limits and paleo-ELA estimates (Allen et al., 2008; Becker et al., 2016; Seguinot et al., 2018; Visnjevic et

al., 2020). LGM precipitation conditions are even more uncertain, with estimates for precipitation anomalies varying widely (around -20 to -60%; e.g., Peyron et al., 1998; Luetscher et al., 2015; Becker et al., 2016), and for which a differential north-south distribution pattern (Florineth and Schluchter, 2000; Becker at al., 2016; Luetscher et al., 2015) is still debated (Seguinot et al., 2018; Visnjevic et al., 2020). Similarly, little is known regarding climatic conditions between the LGM and the YD, besides that significantly lower (>6°C negative anomalies) summer temperatures were still persisting before ca. 15 ka, based

on chironomid and treeline proxies (Heiri et al., 2014a). During the short lived (~1 kyr) YD cooling event, temperatures dropped, with mean annual anomalies varying between 2-3 to 5-9 °C below present day values, depending on the considered proxy between paleoglacial reconstructions (e.g., Protin et al., 2019; Baroni et al., 2021), lacustrine pollen assemblages (Magny et al., 2001) and noble gas speleothem records (Ghadiri et al., 2018; Affolter et al., 2019). On another hand, for the Holocene period, all the available records are in general agreement to indicate that temperatures conditions relatively similar to today

prevailed, with only minor (less than 2°C) deviations (e.g., Davis et al., 2003; Ghadiri et al., 2018; Affolter et al., 2019; Heiri et al., 2014a). Today, there is hence a crucial lack of direct and quantitative *in situ* temperature constraints from within the Alpine massifs during the different reconstructed glacial stages since the LGM.

In this study, we attempt to reconstruct paleotemperatures in the high Alps between the LGM and YD (i.e., Late Glacial) by applying cosmogenic noble gas paleothermometry (Tremblay et al., 2014a). This method exploits the open-system behavior

of cosmogenic $^{3}$He in quartz minerals at Earth surface temperatures (Brook et al., 1993; Shuster and Farley, 2005). Using predictive models of $^{3}$He production and diffusion loss through time and temperature, quantitative constraints on the thermal





history of an exposed rock surface can thus be inferred from the difference between surface-exposure ages derived from the "leaky" $^3$He system and from a cosmogenic nuclide that does not express open-system behavior (e.g., $^{10}$Be), which records a surface-exposure duration assuming non-complex history (Tremblay et al., 2014a, b; 2018). Cosmogenic $^3$He

paleothermometry provides a unique opportunity to obtain quantitative information about past temperature from *in situ* rock surfaces located in the formerly glaciated Alps. Here we explore the applicability of cosmogenic $^3$He paleothermometry along two deglaciation profiles in the northern and western Alps. The advantages of such sampling targets are (1) *a priori* relatively simple exposure history of rock surfaces revealed between the LGM and YD (Wirsig et al., 2016b; Lehmann et al., 2020) with limited shadowing effect (e.g., steep surface, limited vegetation or postglacial sediment cover) and (2) the access to sequences

of gradually exposed but lithologically similar samples, enabling a semi-continuous record of a temperature-change history. Based on $^3$He analytical measurements and forward simulation experiments, we aim to investigate the sensitivity of the *in situ* quartz–$^3$He system in two different high Alpine areas and its suitability for the preservation of a $^3$He thermal signal on Late-Pleistocene timescales. We also compare our results to previous studies applying cosmogenic $^3$He paleothermometry elsewhere in the Alps to gain a further understanding of $^3$He diffusion behavior in quartz at Earth surface temperatures.

## 2 Study sites and sample measurements

### 2.1 Settings and sample collection

Two sites located in major Alpine massifs were selected for this study and have been previously investigated for their deglaciation history: the Mont Blanc Trelaporte (MBTP) profile (Mont Blanc massif, France; Lehmann et al., 2020), located in the western Alps along the western flank of the Mer de Glace valley (NNE exposure); and the SW exposed Gelmersee

(GELM) ridge (Aar massif, Switzerland; Wirsig et al., 2016b), formed by a hanging valley on the east wall of the Haslital valley, in the northern central Alps (Fig. 1, inset). Both sites have steep valley sides that are several hundred meters high with ~30-35° slopes, and are characterized by smoothly-abraded rock surfaces and "roche mountonnée"-like features molded by flowing glaciers. Homogeneous lithologies are exposed along the valley walls, with phenocrystalline granite of the Mont Blanc (Dobmeier, 1998) at the MBTP site, and Aare granite (Labhart, 1977; Abrecht, 1994) as part of the Helvetic crystalline

basement at the GELM site. In both sites, the upper parts of the valley sides are characterized by jagged rock surfaces resulting from active periglacial processes. The transition between smooth and rough surfaces (i.e., the trimline), located at ~2600 m.a.s.l. at the MBTP site and ~2450 m.a.s.l. at the GELM site, either marks the upper limit of the LGM ice surface or of the active (warm-based) eroding glacier layer (i.e., subglacial boundary; Wirsig et al., 2016a).





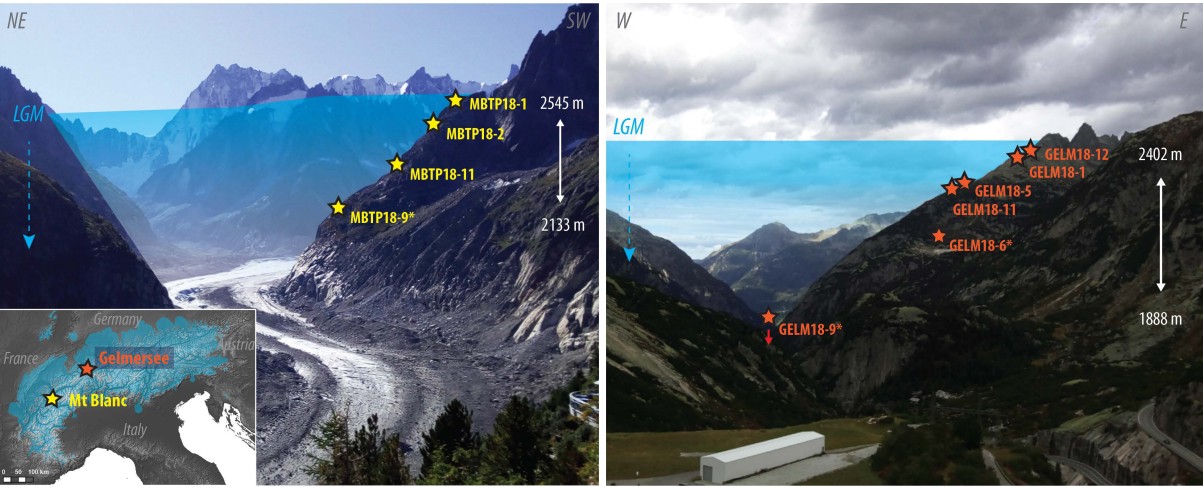

**Figure 1: Mont Blanc Trelaporte (MBTP, left) and Gelmersee (GELM, right) deglaciation profiles since the Last Glacial Maximum (LGM), with the spatial distribution of samples collected for quartz [10]Be (Lehmann et al., 2020; Wirsig et al., 2016b) and [3]He (this study) analyses. Samples with an asterisk have been exposed for ~10-11 kyr (i.e., the entire Holocene period). The inset map indicates the location of the two study sites within the European Alps and the extent of ice cover during the LGM (in blue; Ehlers et al., 2011).**

Ice-surface lowering of around 400 (MBTP) to >500 (GELM) meters between the LGM to the YD has been recorded using *in*

*situ* [10]Be cosmogenic exposure dating on bedrock surfaces collected at regular intervals along each profile, starting from just

below the trimline (Figs. 1-2, Table 1; Lehmann et al., 2020; Wirsig et al., 2016b). In this study, samples were collected again

for [3]He experiments from the exact same locations previously collected for [10]Be dating by Lehmann et al. (2020; MBTP profile,

samples MBTP18 -1, -2, -11 and -9, n=4) and Wirsig et al. (2016b; GELM profile, samples GELM18 -12, -1, -5, -11, -6 and -

9, n=6; Fig. 1, Table 1). All samples are from glacially scoured bedrock surfaces, except GELM18-11, which comes from the

top of a ~5-m high boulder of similar lithology deposited during the post-LGM ice-surface lowering (Wirsig et al., 2016b).




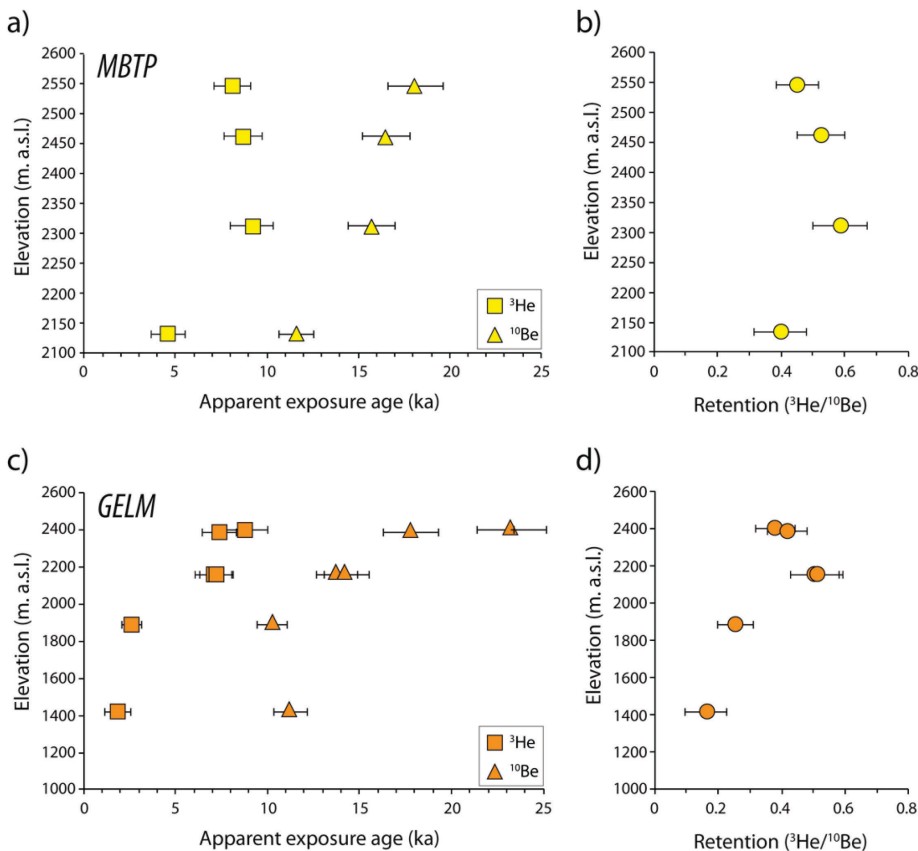

**Figure 2: Apparent ³He (this study) and ¹⁰Be (Lehmann et al., 2020; Wirsig et al., 2016) exposure ages (a-c), and ³He/¹⁰Be exposure age ratios or retention (b-d) as a function of elevation along the two deglaciation profiles (MBTP: a-b; GELM: c-d).**

**Table 1: MBTP and GELM sample information.**

| Profile | Sample | Lat./Long. (°N/°E) | Alt. (m. asl) | ¹⁰Be age (ka)[2] | ³He age (ka)[2] | MARST (°C)[3] | Modern EDT (°C)[4] | Paleo IsoEDT (°C) |
|---|---|---|---|---|---|---|---|---|
| MBTP | MBTP18-1 | 45.9083/6.9311 | 2545 | 18.1±1.5 | 8.1±1.0 | 1.2 | 5.8 | 3±1.5 |
| | MBTP18-2 | 45.9086/6.9319 | 2460 | 16.5±1.3 | 8.7±1.0 | 1.7 | 6.2 | 0.5±2 |
| | MBTP18-9[1] | 45.9124/6.933 | 2133 | 11.6±1.0 | 4.6±0.9 | 2.4 | 8.0 | 8±2.5 |
| | MBTP18-11 | 45.9108/6.9315 | 2310 | 15.7±1.3 | 9.2±1.1 | 3.3 | 7.0 | -1.5±2.5 |
| GELM | GELM18-1[1] | 46.6218/8.3257 | 2387 | 17.8±1.5 | 7.4±0.9 | 3.1 | 7.9 | -5.5±3 |
| | GELM18-5 | 46.6185/8.3215 | 2155 | 13.8±1.1 | 7.0±1.0 | 4.3 | 9.1 | -11±3 |
| | GELM18-6 | 46.6151/8.3212 | 1888 | 10.2±0.8 | 2.6±0.5 | 5.7 | 10.6 | 9.5±3 |
| | GELM18-9 | 46.6136/8.3071 | 1418 | 11.2±0.9 | 1.8±0.7 | 8.1 | 13.1 | 14.5±4 |
| | GELM18-11 | 46.618/8.3217 | 2154 | 14.3±1.2 | 7.2±0.9 | 4.3 | 9.1 | -11±3 |
| | GELM18-12 | 46.6221/8.3258 | 2402 | 23.3±1.9 | 8.8±1.2 | 3.0 | 7.8 | -4.5±3 |

[1]Samples used for ³He diffusion experiments. [2] Re-calculated ¹⁰Be (after Wirsig et al., 2016 and Lehmann et al., 2019) and calculated ³He exposure ages using the non-time dependent scaling scheme of Stone (2000; Balco et al., 2008), using SLHL production rates of



4.01 at.g⁻¹.yr⁻¹ ($^{10}$Be; Borchers et al., 2016) and of 116 at.g⁻¹.yr⁻¹ ($^{3}$He; Vermeesch et al., 2009) and assuming a rock density of 2.65 g.cm⁻³. See the supplementary material for the details of $^{10}$Be and $^{3}$He concentrations (Table S3). [3]Estimated Mean Annual Rock Surface Temperature at ~3 cm depth. [4]Modern EDTs calculated using $E_a$ of 93.5 (MBTP) or 98.5 (GELM) kJ.mol⁻¹, samples specific MARST estimates and using 10°C annual and 5°C diurnal amplitudes.

## 2.2 Samples preparation and measurement experiments

Aside from the exposure time of a rock surface determined using independent chronometers (in our case *in situ* $^{10}$Be surface-exposure dating), cosmogenic $^{3}$He paleothermometry requires at least two additional pieces of information. First, predictive models of thermally-activated $^{3}$He diffusion rely on quartz sample-specific $^{3}$He diffusion kinetics parameters (i.e., activation energy $E_a$ and the diffusion at infinite temperature scaled to the diffusion length scale (pre-exponential factor) $D_0/a^2$; Tremblay et al., 2014a, b), which need to be experimentally determined. Second, measurement of the total natural cosmogenic $^{3}$He accumulated in the quartz sample permits us to estimate the loss by diffusion which occurred throughout the exposure time of the rock surface.

Rock samples were disaggregated using a high-voltage pulse-based system (SELFRAG equipment, Institute of Geological Sciences, University of Bern) to optimize the breaking of the rock along crystal grain boundaries. After rinsing, quartz mineral grains were separated from the other minerals (heavy minerals and feldspar) by magnetic separation and froth flotation (e.g., Nichols and Goehring, 2019) and were additionally etched in 1% HF for 3 weeks at room temperature to ensure the removal of any adhesive micro-mineral particles which may contaminate the $^{3}$He measurements. For both sites, grain-size distribution after removal of the finer fraction (<200 μm) is centered around 850 μm diameter (Fig. S1). One representative sample per profile was selected for diffusion kinetics experiments (MBTP18-9 and GELM18-1). For these samples, 200 to 300 mg of quartz grains were visually selected under a binocular microscope to avoid obvious mineral inclusions and sent to the Francis H. Burr Proton Therapy Center (Massachusetts General Hospital) for proton beam irradiation (Shuster et al., 2004; Shuster and Farley, 2005) in February 2019. After several months of rest to lower the level of radioactivity, one individual coarse quartz grain (~700 μm diameter for MBTP18-9 and ~900 μm diameter for GELM18-1, based on calibrated petrographic microscope measurements) with no obvious fractures and no mineral or fluid inclusion was selected from each irradiated sample to conduct step-degassing experiments. For natural $^{3}$He measurements, the 800-1000 μm quartz grain fraction (i.e., 400-500 μm radii) from each sample was isolated, as we anticipated this fraction would have best preservation potential of a measurable $^{3}$He signal for the expected range of thermal histories experienced by the MBTP and GELM samples (Brook and Kurz, 1993; Tremblay et al., 2014a). Three replicates per sample were prepared in tantalum packets containing ~100 mg of quartz grains for analysis of natural $^{3}$He concentrations.

Both stepwise-heating experiments to characterize the $^{3}$He diffusion kinetics in the proton-irradiated quartz grains and bulk degassing measurements to determine the natural cosmogenic $^{3}$He abundances in the ~100 mg quartz grain replicates were carried out at the BGC Noble Gas Thermochronometry Lab (Berkeley, USA). The measurements were conducted with an MAP 215-50 sector field mass spectrometer following similar procedure to Tremblay et al. (2014b). For diffusion kinetics experiments, samples were heated over thirty to forty heating steps lasting 0.5 to 4 hours, with temperatures increased from





100 up to 550°C and including at least one retrograde heating cycle (Tables S1-S2). Blank measurements at room temperature were regularly conducted throughout the experiments for background subtraction from the measured raw signals, with averaged ${}^3$He blank corrections of 2.1 x10$^4$ atoms (MBTP18-9) and 4.9 x10$^4$ atoms (GELM18-1). For natural cosmogenic ${}^3$He
measurements (Tables S3-S4), each tantalum packet was heated in two, 15-minute long heating steps at 800 and 1100 °C, with no gas release observed in the second step. Hot blanks on empty tantalum packets were measured, from which an averaged ${}^3$He blank correction of 7.7 x 10$^3$ atoms was applied.

## 3 Analytics approach: constraining diffusion kinetics and Effective Diffusion Temperature

In this study, we used Matlab codes initially developed by Tremblay et al. (2014a, b; 2018; 2021, code available on Zenodo at
https://doi.org/10.5281/zenodo.5808021) to (1) determine ${}^3$He diffusion kinetics from step-heating experiment data applying a multi-diffusion domain (MDD; Lovera et al., 1989, 1991) model framework, and (2) numerically simulate ${}^3$He loss for different thermal histories using the sample-specific diffusion kinetics information. The predictive model of ${}^3$He diffusion with time also includes ${}^3$He production by cosmic ray incidence using a ${}^3$He production rate in quartz at sea level and high latitude (SLHL) of 116 at.g$^{-1}$.yr$^{-1}$ (Vermeesch et al., 2009), scaled to the sample geographic location and elevation according to the
non-time dependent scaling scheme of Stone (2000; Balco et al., 2008). For consistency, apparent ${}^3$He and ${}^{10}$Be exposure ages along the deglaciation profiles are (re-)calculated following the same approach (with SLHL ${}^{10}$Be production rate of 4.01 at.g$^{-1}$.yr$^{-1}$; Borchers et al., 2016), using the measured ${}^3$He (this study) and literature ${}^{10}$Be concentrations (previous studies, Wirsig et al., 2016b; Lehmann et al., 2020), assuming negligible erosion (Fig. 2; Tables 1 and S3). Recalculated ${}^{10}$Be exposure ages define the exposure time of sampled rock surfaces during which we simulate ${}^3$He production and diffusion as a function of
ambient temperature. In order to account for periodic temperature oscillations (e.g., diurnal, seasonal, geological), effective diffusion temperatures (EDTs, Tremblay et al., 2014a) are used as temperature inputs in the predictive ${}^3$He diffusion model as detailed below.

### 3.1 Effective Diffusion Temperature estimates

Rock surfaces experience temperature fluctuations at the diurnal, seasonal and longer timescales, which will all activate
thermal diffusion of ${}^3$He in quartz (Tremblay et al., 2014a). Because ${}^3$He diffusivity increases exponentially with temperature, a constant model temperature required to explain a total ${}^3$He loss (i.e., corresponding to the mean diffusivity through time) from a geological sample will equal or exceed the actual mean temperature experienced at the rock surface. This temperature is called Effective Diffusion Temperature (EDT; Christodoulides et al., 1971; Tremblay et al., 2014a), and is function of the ${}^3$He diffusion activation energy $E_a$, the long-term mean (rock surface) temperature and the diurnal and seasonal temperature
amplitudes.

In our approach, temperature variables used for the EDT calculation at the different sampling sites (which are subsequently used for ${}^3$He diffusion simulations; cf. Sect. 3) were estimated as follows. Mean annual air temperatures (MAATs) at each



sampling sites along the MBTP and the GELM profiles were calculated by linear interpolation assuming a lapse rate of 5°C/km

(Gramiger et al., 2018) based on mean annual temperatures recorded by nearby reference weather stations at Chamonix (1042

m.a.s.l., ~5 km west; period 1993-2012; Magnin et al., 2015a) and Grimsel-Hospiz (1980 m.a.s.l; ~5 km south; period 2010-

2020, data MeteoSwiss), respectively. Mean Annual Rock Surface Temperatures (MARSTs) are typically higher than MAATs,

with the difference amplified between south- and north-exposed slopes (Gruber et al., 2003). Boeckli et al. (2012a), based on

57 sensor measurements on snow-free rock slopes >55°, showed that the measured difference between MAAT and MARST

increased linearly from <1°C to up to 10°C depending on potential incoming solar radiation (PISR), which is largely controlled

by rock surface aspect and angle, in addition to elevation. For moderately inclined surfaces, the difference between MARST

and MAAT is expected to be reduced by ~1-3°C due to micro-topography and snow-insolating effects (Hasler et al., 2011).

To estimate MARSTs, we calculated the PISR at each sampling site using the Area Solar Radiation tool (ArcGIS software,

version 10.3.1) applied to a 30 m resolution Digital Elevation Model (SRTM 1 Arc-Second data) at the study sites. The

calculation was performed at hourly resolution using data from one year (2000), assuming no nebulosity and using a sky size

of 512 cells (Magnin et al., 2015a; Mair et al., 2020). Based on the linear relationship between MAAT-MARST and PISR

from Boeckli et al. (2012a), we estimated the average MARST-MAAT difference assuming snow-free conditions at each site,

to which we then subtracted 2 °C to take into account snow-insulating and micro-topographic effects in moderately steep

terrain (Hasler et al., 2011). Final differences between MAAT and MARST of +1°C and +2.5°C were thus obtained for the

north-exposed MBTP and the southwest-exposed GELM sites, respectively. These estimates are consistent with *in situ* MAAT

and MARST measurements available in nearby areas with similar orientations, elevations and slope inclinations (e.g., Gruber

et al., 2004; Magnin et al., 2015a, b; Haberkorn et al., 2017; Gramiger et al., 2018; Guralnik et al., 2018), and were thus

to estimate the MARSTs at each sampling sites.

A mean annual temperature amplitude of 10°C and diurnal amplitude of 5°C were adopted for the two sites, based on long-

term (i.e., several years) temperature records from the Chamonix and Grimsel-Hospiz weather stations, and from direct *in situ*

rock surface measurements available in the Alps (Gruber et al., 2004; Magnin et al., 2015b; Gramiger et al., 2018; Mair et al.,

2020; Guralnik et al., 2018). These estimates are consistent with the annual/diurnal amplitudes obtained from the spatially-

interpolated land surface climate data set WorldClim 2.0, based on gridded time series of meteorological data from available

weather stations (target temporal range 1970-2000; 1-km resolution; Fick and Hijmans, 2017).

## 3.2 Diffusion kinetics determination

Diffusion kinetics parameters ($E_a$ and $\ln(D_0/a^2)$) were determined following a multi-step procedure. For each proton-irradiated

sample (one per site), we first produced an Arrhenius plot displaying the natural log of diffusivity $D$ (scaled to the diffusion

length scale $a$) as a function of inverse temperature (Fig. 3), calculated from each [3]He degassing step experiment using the

equation of Fechtig and Kalbitzer (1966; in Tremblay al., 2014b).

Preliminary tests using the MDD model framework described by Tremblay et al. (2014b) were carried out to determine the $E_a$

required to best fit the Arrhenius data points in the lower temperature range (~70 to 100°C) assuming a single diffusion array,



as well as the (minimum) number of diffusion domains to explain the entire data set (i.e., all heating steps). Iterative experiments using the MDD model with the minimum number of domains inferred from the preliminary tests were then conducted for a range of increasing $E_a$ up to 100 kJ/mol (with 0.5 increment; Fig. 3), based on existing $E_a$ estimates reported for quartz in the literature (Tremblay et al., 2014b; Tremblay et al., 2018). In each experiment, $E_a$ was kept common to each

domain (Lovera et al., 1991; Baxter et al., 2010) while $\ln(D_0/a^2)$ and gas fraction were allowed to vary between the different domains until the misfit coefficient was minimized between the simulated and observed $\ln(D/a^2)$ values for all the heating steps.

Because heating step degassing experiments conducted in laboratory do not permit to capture ³He diffusion behavior at Earth's surface temperature range (i.e., from around -30 to 30°C), we next introduced an extra calibration step using the measured

natural ³He concentration from the samples with Holocene-only exposure (from both GELM and MBTP sites) to constrain diffusion kinetics which might be more representative in the range of temperature conditions experienced by Alpine rock surfaces. Based on independent global and regional paleoclimate proxy records, samples exposed during the Holocene have experienced relatively stable averaged temperature conditions with only minor variations (i.e., less than 2°C; cf. Sect. 1). Assuming no complex exposure history, the ³He signal recorded in these samples can therefore be considered representative

of ³He diffusion occurring at a constant temperature equivalent to the EDT calculated at each sample site. We thus conducted another series of numerical experiments to test sets of diffusion kinetic parameters determined from our laboratory experiments that explain the natural ³He concentrations recorded in the Holocene samples available in our two study sites (MBTP18-9, ¹⁰Be exposure age of ca. 11.6 ka; GELM18-9 and-6, ¹⁰Be exposure ages of ca. 10.2 and 11.2 ka). For each study site, the corresponding set of $E_a$ and associated $\ln(D_0/a^2)$ per domain (with gas fraction) was thus implemented in the forward simulation

model of ³He evolution with time and temperature (Tremblay et al., 2014a), which was run over a time period equivalent to the recalculated ¹⁰Be exposure age and for a constant temperature equivalent to the modern EDT (recalculated accordingly for each $E_a$, based on sample specific temperature variables, cf. Sect. 3.1.) of the Holocene sample(s). Diffusion kinetics parameters for which resulting modeled ³He concentration matched within error the observed natural one (i.e., from the Holocene sample considered) were retained, with the solution producing the best match considered as the final calibrated

diffusion kinetics parameters (Fig. 3). We assume the Holocene-calibrated parameters apply to all the samples (i.e., including with longer exposure durations) collected along each profile (MBTP or GELM), given the homogenous lithology between samples, and were used as the default diffusion kinetics parameters for the numerical experiments conducted in the next sections.





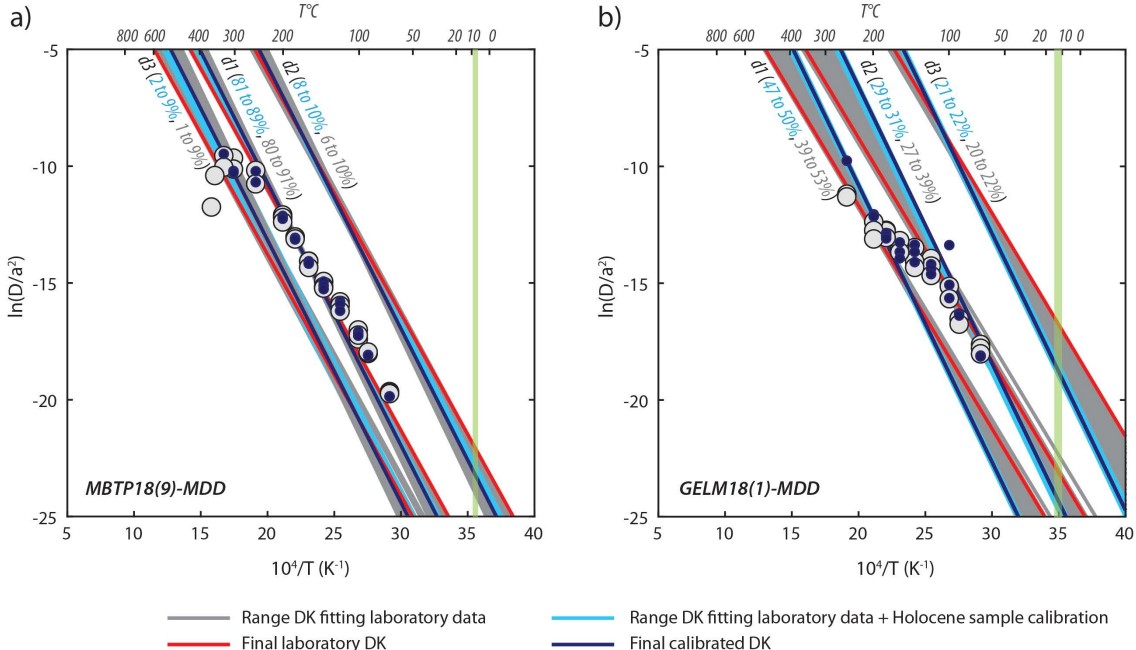

**Figure 3: Arrhenius plots of ³He step-degassing experiments (grey circles) conducted on one representative sample per study site. Final diffusion kinetics parameters were determined using a multi-step procedure, including the determination of MDD diffusion kinetics (DK) parameters that (1) best fit the laboratory data for a range of increasing $E_a$ (79 (a) and 86 (b) to 100 kJ/mol; grey lines; red lines show final laboratory parameters providing the best match in the lower temperature range), and (2) match within uncertainties the natural ³He concentration recorded in the Holocene calibration samples (light blue lines). The dark blue line indicates the final calibrated DK parameters best matching the step-degassing experiments and the natural Holocene ³He concentration (dark dots represent the corresponding $\ln(D/a^2)$ values modelled along the heating experiment schedule). For the MBTP site (a), sample MBTP18-9 was used for both the laboratory step-degassing experiment as well as the Holocene calibrating sample. For the GELM site (b), sample GELM18-1 was used for the laboratory step-degassing experiment, while sample GELM18-6 and -9 were used for Holocene calibration. The light green line indicates the EDT range associated with the Holocene calibration sample(s). The gas fraction assigned to each domain for both laboratory (grey font) and Holocene-calibrated (blue font) DK parameters is also indicated along lines.**

## 4 Results

### 4.1 Diffusion kinetics and sensitivity tests

We present in Figure 3 the range of diffusion kinetics parameters ($E_a$ and $\ln(D_0/a^2)$) fitting the laboratory degassing experiments (one representative sample for each site), and which in addition predict the observed natural ³He concentrations from the Holocene calibration samples for a constant EDT equivalent to the modern EDT. Degassing experiment data indicate relatively first-order Arrhenius behavior for quartz ³He diffusion of MBTP18-9, with one dominant linear array accounting for ~85% of ³He release (Fig. 3a, Table 2). The remaining ~15% gas fraction is distributed within two additional minor diffusion domains, one of higher retentivity and one of lower retentivity (Tremblay et al., 2014b). GELM18-1 exhibits more complex quartz ³He diffusion behavior, with gas release distributed more equally (gas fraction between 20 to 50%) within three linear arrays (Fig.



3b, Table 2), which can be interpreted as three distinct diffusion domains with each domain contributing significantly to [3]He retention over geological times.

**Table 2: Diffusion kinetics parameters for MBTP and GELM sites.**

| Profile | [2]Range of Holocene-calibrated parameters | | | | [3]Final Holocene-calibrated parameters | | | [4]Final laboratory parameters | | |
|---|---|---|---|---|---|---|---|---|---|---|
| | Ea (kJ/mol) | n domain | $\ln D_0 a^2$ (ln(s⁻¹)) | Gas fraction (%) | Ea (kJ/mol) | $\ln D_0 a^2$ (ln(s⁻¹)) | Gas fraction (%) | Ea (kJ/mol) | $\ln D_0 a^2$ (ln(s⁻¹)) | Gas fraction (%) |
| [1]MBTP | 91.5 to 96 | d1 | 11.11 to 12.56 | 81 to 89 | 93.5 | 11.78 | 85 | 85.9 | 9.67 | 93 |
| | | d2 | 16.11 to 17.77 | 8 to 10 | | 16.78 | 9 | | 14.67 | 6 |
| | | d3 | 8.67 to10 | 2 to 9 | | 9.33 | 6 | | 6.89 | 1 |
| [1]GELM | 96.5 to 100 | d1 | 12.22 to 13.33 | 47 to 50 | 98.5 | 12.89 | 50 | 79.5 | 7.44 | 43 |
| | | d2 | 16.33 to 17.56 | 29 to 31 | | 17.11 | 29 | | 10.33 | 36 |
| | | d3 | 22.11 to 23.11 | 21 to 22 | | 22.67 | 21 | | 16.67 | 21 |

[1]Diffusion kinetics measurements made on one representative sample per profile: MBTP18-9 (350 µm spherical equivalent radius) and GELM18-1 (450 µm spherical equivalent radius). [2]Range of MDD diffusion kinetics parameters obtained by fitting laboratory experimental data and matching [3]He concentrations (within 1σ error) from Holocene calibration samples. [3]Best-fitting MDD diffusion kinetics parameters obtained by fitting laboratory experimental data matching [3]He concentrations from Holocene calibration samples. [4]MDD diffusion kinetics parameters based only on laboratory experimental data, and providing the best match in the lower temperature range of the heating schedule (~70-100 °C).

In order to explore the theoretical sensitivity (and potential variability) of the MBTP and GELM quartz, we numerically evaluated the time required for the concentration of [3]He in each sample to reach steady-state (i.e., thermal loss balanced with cosmic-ray induced production gain) as function of constant EDT. Forward simulations using the [3]He diffusion predictive model of Tremblay et al. (2014a, b) and the final Holocene-calibrated diffusion kinetic parameters were thus run for a range of isotherms representative of Earth surface EDTs (hereafter referred to as isoEDTs; tested range from -30 to 30°C), assuming a 450µm radii and no initial [3]He concentration. Equilibrium conditions were assumed to be reached once no significant change in [3]He concentration was recorded (<1% per kyr). While we observe some variability in [3]He diffusion behavior and derived diffusion kinetics parameters between MBTP and GELM quartz (Fig. 3, Table 2), results from sensitivity tests in terms of steady-state times are relatively similar. For isoEDTs between -10 and 10°C, bracketing approximately potential EDTs values experienced along both deglaciation profiles between the LGM and today, the time predicted for [3]He diffusion to reach equilibrium varies between ~10 kyr (isoEDT of 10°C) and ~20 kyr (isoEDT of -10°C; Fig. 4). Interestingly, while steady-state time estimates remain relatively constant for quartz from both sites at ca. 20 kyr for colder isoEDTs (-10 to -30°C), we observe a pronounced non-linear dependence for EDTs above 0°C, resulting in much shorter equilibrium times in the high EDTs range (less than 5 kyr for EDT above 20 °C, Fig. 4).

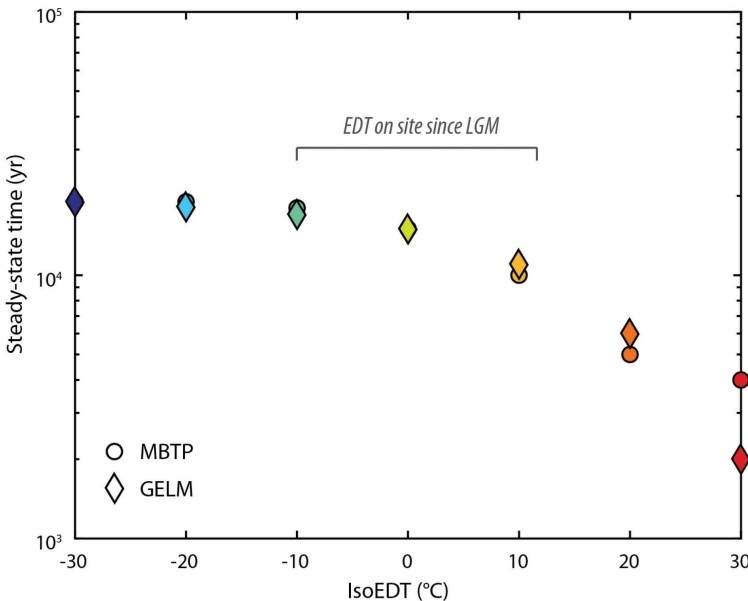

**Figure 4: Theoretical [3]He steady-state time estimates for isoEDTs varying between -30 to 30 °C, using the final Holocene-calibrated diffusion kinetics parameters determined for each study site, and assuming 450 μm grain radius.**

## 4.2 [3]He exposure ages and PaleoIsoEDTs

For each site, apparent [3]He exposure ages are systematically lower (from 20 to 75%) than apparent [10]Be exposure ages (Table 1, Fig. 2). While [10]Be ages show a general decrease with elevation, in agreement with progressive ice thinning along a

deglaciation profile in the high Alps during the Late Glacial, this trend is less evident for apparent [3]He ages (Fig. 2). The evolution of retention ([3]He/[10]Be exposure ages ratio) with elevation differs between the two sites, with an apparent decrease in retention for low-elevation/younger samples along the GELM profile (~1500 to 2500 m a.s.l.), which is not visible along the MBTP profile (similar retention between MBTP samples), which is more restricted in elevation range (~2100 to 2600 m a.s.l.; Fig. 2).

To determine the apparent constant EDT (that we refer to as paleoIsoEDT) from the natural [3]He signal recorded in each sample, [3]He diffusion predictive models (implemented with the final Holocene-calibrated diffusion kinetics) were run for a time period equal to the sample's [10]Be exposure age and for a range of isoEDT (isothermal holding between -10 to 15 °C for MBTP; -25 to 20°C for GELM, 1°C increment). The isoEDT leading to best-matching synthetic [3]He concentration with the observed natural [3]He concentration was retained as the paleoIsoEDT (Fig. 5, Table 1). As Holocene samples were used to calibrate the

diffusion kinetics (cf. Sect. 4.1), it is expected that [3]He derived paleoIsoEDTs from theses samples are equivalent to their respective modern EDTs. On the other hand, all pre-Holocene samples at both sites have paleoIsoEDTs that are lower than their corresponding modern EDTs (Fig. 5a, d; Table 1). For the MBTP profile, the difference between modern EDTs and





paleoIsoEDTs varies from around 3 to 9°C. This difference is even greater for the GELM profile, where paleoIsoEDTs are around 12 to 20°C lower than their associated modern EDTs. Pre-Holocene samples are located well above (200 to 500 meters)

Holocene samples, and all above 2000 m a.s.l. While paleoIsoEDTs derived from the high-elevation/pre-Holocene samples agree within error for each site, they clearly depart from EDTs obtained from the low-elevation/Holocene sample(s), by ~6° (MBTP site) and ~18°C (GELM site) based on the obtained bimodal distributions (Fig. 5b, e). After correcting for temperature decrease with elevation (assuming a lapse rate of 5°C/km), the difference between pre- and Holocene samples paleoIsoEDTs is still significant for GELM (~10 to 20°C, Fig. 5f). For MBTP, although elevation-corrected paleoIsoEDTs from two high-

elevation/pre-Holocene samples (MBTP18-2 and -11) are still clearly distinguishable from the low-elevation/Holocene sample (MBTP18-9; Fig. c), the probability distribution appears closer to unimodal since the paleoIsoEDT from the highest sample (MBTP18-1) partially overlap with MBTP18-9.

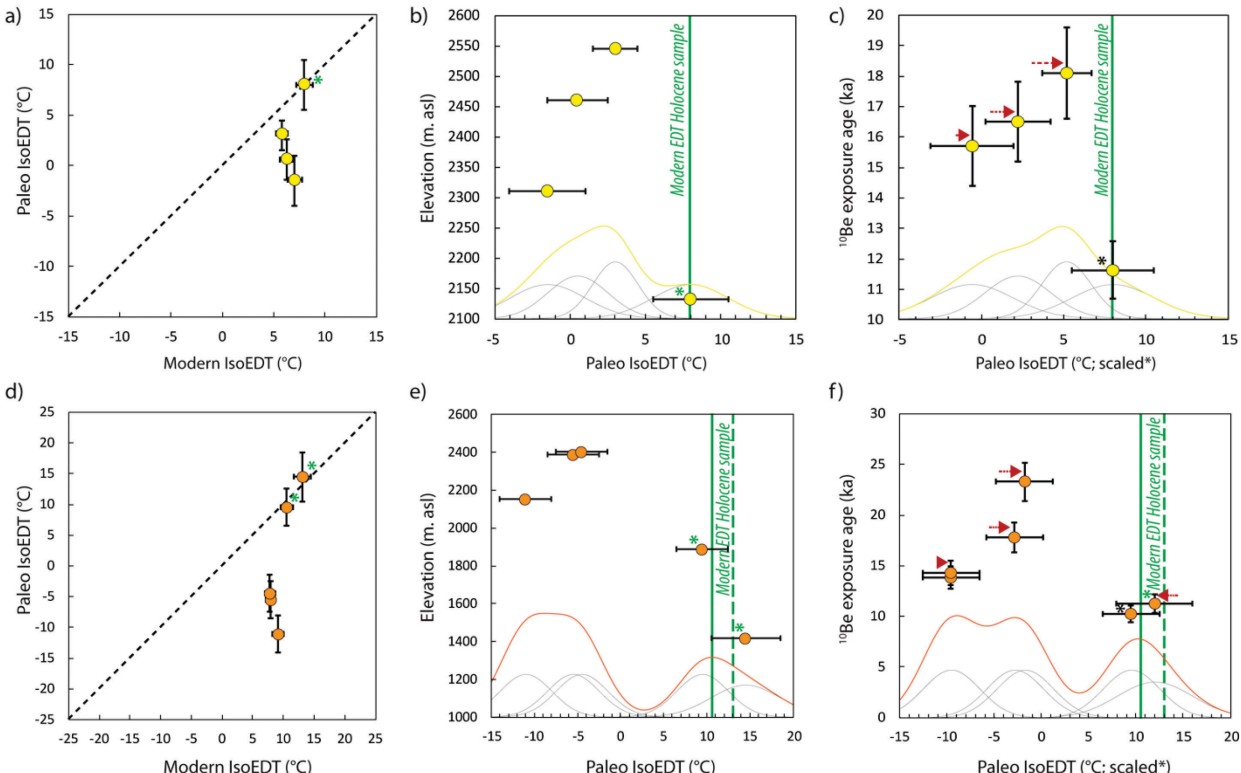

**Figure 5: Distribution of ³He derived paleoIsoEDTs along the MBTP (a-c, top) and GELM (d-f, bottom) deglaciation profiles.**
**Holocene samples used for calibration are marked by an asterisk. (a,d) Paleo-IsoEDTs relative to modern EDTs (black dashed line is 1:1); (b,e) relationship between paleoIsoEDT and elevation, and (c,f) relationship between paleoIsoEDT and ¹⁰Be exposure age, after correction for lapse rate (marked by red arrows, relatively to Holocene sample marked with a black asterisk). Green (dashed) lines are modern EDTs for Holocene samples. The thin lines represent the sum (yellow/orange) of the individual (grey) probability distribution of paleoIsoEDTs.**





## 4.3 Forward simulations with time-varying EDT

Based on global and regional paleoenvironmental records, we can expect that pre-Holocene samples collected at MBTP and GELM sites have experienced at least one main significant temperature change, marking the transition from (colder) Late Glacial to warmer and more stable Holocene conditions (cf. Sect. 1 for details).

Following this observation, we first investigate the theoretical time needed for the MBTP and GELM $^3$He quartz systems to re-adjust to a change in temperature in a warming scenario, assuming that these systems were already at steady-state conditions. Forward simulations (450 µm radii assumed) were run for different time-EDT scenarios involving an initial EDT (ranging from -30 to 30°C; initial $^3$He concentration at steady-state with initial EDT) followed by a step warming event (+2, +5, +10 +15, +20, +25, +30°C; over 0.1 to 1 kyr depending of the sensitivity of the quartz system for the considered EDT scenario), after which resulting warmer EDT was maintained until full re-adjustment of the $^3$He-quartz system. We considered full re-adjustment to have occurred when modelled $^3$He concentrations following the step warming event match within 10% the $^3$He concentrations expected for an isoEDT equivalent to the final (warmer) EDT (Fig. 6a). We present simulation results in Figures 6b-c. For past EDTs <0°C followed by a step warming up to 20°C, readjustment times are all longer than 10 kyr, either considering MBTP or GELM diffusion kinetics. Estimates of LGM-temperature anomalies suggested for the European Alps are equivalent to an apparent warming of 5 to 15 °C (cf. Sect. 1). When considering EDT scenarios with similar warming range applied to our study sites with modern EDTs around 5 to 10°C (at pre-Holocene sampling sites; i.e., equivalent to initial past EDT between 0 to 5°C and -10 to -5°C for 5 and 15°C warming step, respectively), our simulation outcomes show relatively long re-adjustment times from around 20 to 45 kyr (Fig. 6). We should note however that these times are maximum estimates since we considered $^3$He quartz systems at steady-state conditions with initial cold EDTs before the warming event.

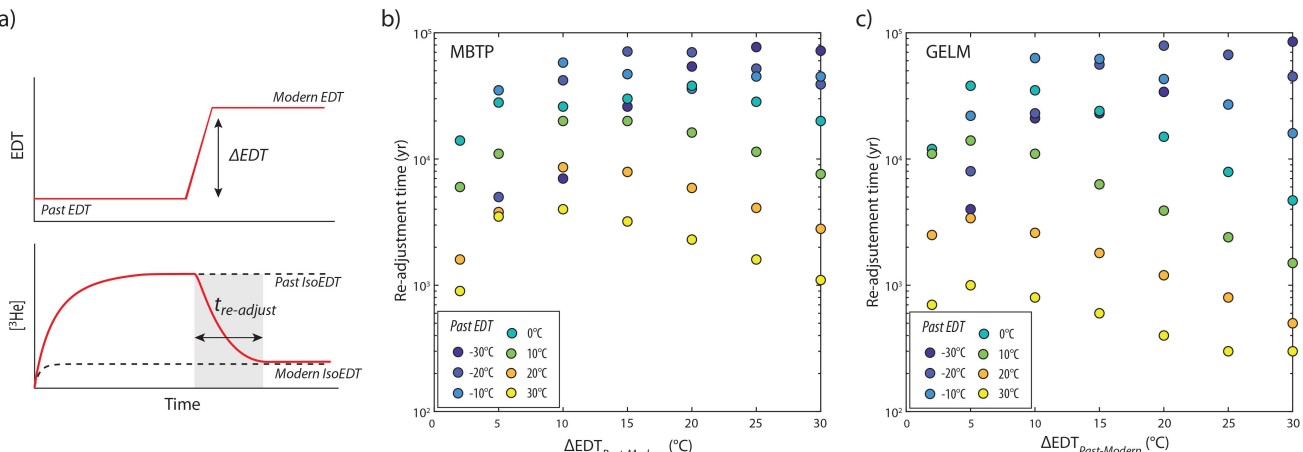

**Figure 6: Conceptual approach (a) and output results for MBTP (b) and GELM (c) of $^3$He re-adjustment time ($t_{re-adjust}$) for one-step EDT change scenarios (temperature warming from 2 to 30°C), using the final diffusion kinetics parameters determined for each study site, and assuming 450 µm grain radius. Calculations assume that $^3$He concentrations were already at steady-state for past EDT conditions (i.e., as would be expected for infinite exposure time) prior to imposing the temperature change.**





In a second set of experiments, we explore the thermal memory of $^3$He in quartz for a step warming EDT scenario fixed in
time that is more representative of the post-LGM paleoclimate history in the Alps, including: (1) an initial cold period starting
at 24 ka with an imposed EDT set 15°C lower compared to modern EDT (maximum LGM temperatures anomalies, cf. Sect.
1); (2) a warming step to modern EDT that is either progressive from 24 to 10 ka (i.e., consistent with a Younger Dryas-
Holocene transition) or abrupt between 11 and 10 ka, and (3) stable conditions at the modern EDT throughout the Holocene
(last 10 kyr, Fig. 7a). Simulations of $^3$He diffusion and concentration evolution were conducted for each pre-Holocene sample
following this scenario, with the time period and the time-dependent EDT variable set accordingly to each sample $^{10}$Be
exposure age and modern EDT, respectively. For all GELM and MBTP samples, these forward simulations result in synthetic
$^3$He concentrations significantly lower than their respective measured $^3$He concentrations. In Figure 7b we present the results
for sample MBTP18-1, for which we observed the smallest difference between modern EDT and paleoIsoEDT (Fig. 5a; Table
1).

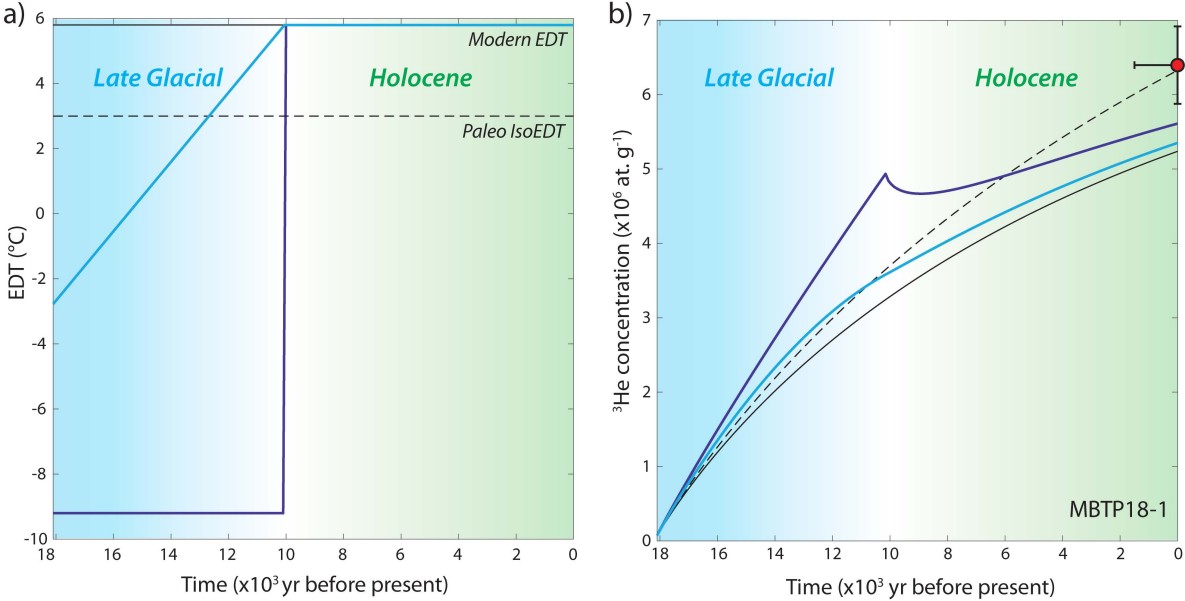


**Figure 7: a) Simplified warming EDT scenario since the LGM (~24 ka), with progressive and abrupt EDT changes in light and dark blue lines, respectively; b) Synthetic evolution of $^3$He concentration (blue lines) compared with the natural $^3$He concentration recorded in MBTP18-1 (red circle). The $^3$He concentration evolution is also indicated for a constant-temperature scenario at the modern EDT and paleoIsoEDT (set in Figure 5; black solid and dashed lines, respectively). We were unable to reproduce the**
**observed natural $^3$He concentration for any samples with pre-Holocene exposure under this simplified LGM EDT scenario.**

To further investigate potential effects of a larger EDT difference between modern and past conditions, and/or a more recent
EDT change, we performed an additional set of simulations using step warming EDT scenarios with more free parameters.
Scenarios with an EDT change occurring from $10^4$ to $10^2$ years ago and with difference between past and modern EDTs
(ΔEDT) up to 40°C were tested iteratively on each pre-Holocene sample, assuming no initial $^3$He concentration and with total
exposure time and EDT variables adjusted accordingly, as described above. Scenarios for which we could reproduce the




observed natural [3]He concentration (within uncertainties) were accepted, resulting in a range of different possible scenarios with varying ΔEDT and time of EDT change for each pre-Holocene sample (Fig. 8). For both sites, we observed a similar pattern between ΔEDT and time of EDT change: the further back in time the EDT change occurs, the greater the ΔEDT that is needed to reproduce observed natural [3]He concentrations. In addition, for any given time of EDT change, ΔEDTs tend to be

inversely correlated with sample elevation/[10]Be exposure age. Within these similarities, the two sites differ by the magnitude of the ΔEDT required to reproduce observed natural [3]He concentrations. For example, along the MBTP site (Fig. 8a), ΔEDTs of 5 ºC occurring a few kyr ago are required to explain [3]He concentrations measured from the highest/oldest sample (MBTP18-1), while ΔEDTs of 35°C occurring a few kyr ago are required to explain [3]He concentrations measured from the lowest /youngest pre-Holocene sample (MBTP18-11). For the same sites, ΔEDTs of 3 and 15°C are required if the ΔEDT occurred

within the last centuries. On another hand, for the GELM site, our simulations found no ΔEDT solution if the EDT change is applied prior to 1 ka (within our ΔEDT limit of 40°C; except for GELM18-12; Fig. 8b). In the case of EDT change occurring within the last centuries, ΔEDTs for the GELM samples are significantly larger than for MBTP samples, with ΔEDTs between 15 to >30°C required for the highest/oldest samples (GELM18-12 and -1). For the intermediate samples (GELM18-11 and -5) which are also exhibiting the greatest [3]He-[10]Be age differences, numerical solutions could only be recovered for very recent

EDT changes (≤200 yr) and with ΔEDT >35°C.

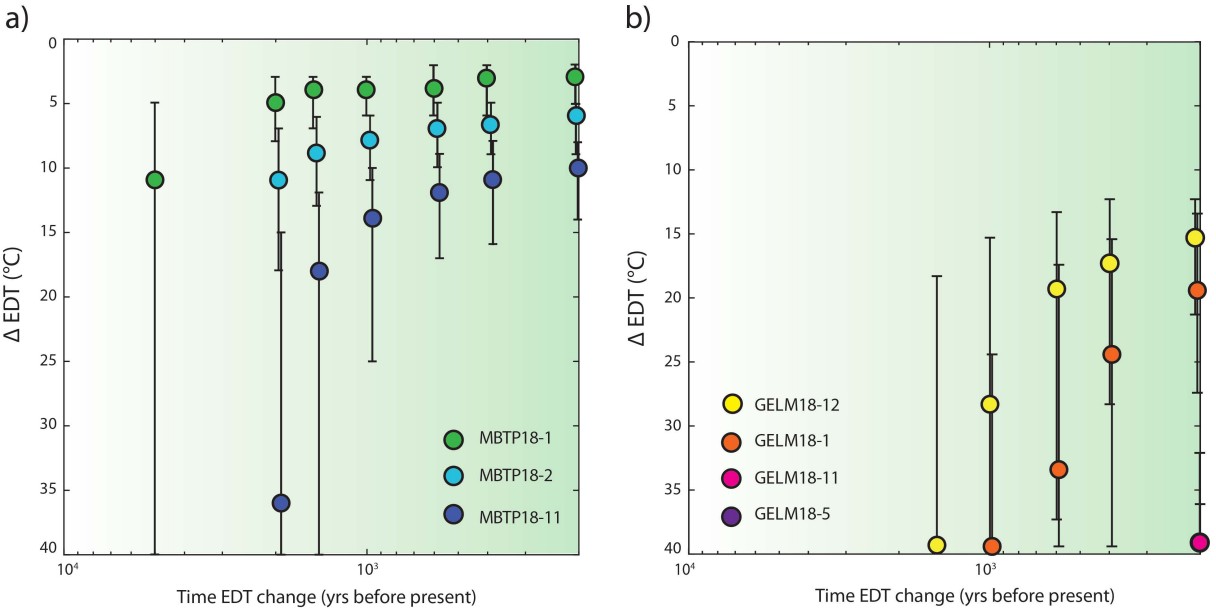

**Figure 8: One-step EDT change scenarios that reproduce the observed natural [3]He concentration for each pre-Holocene MBTP (a) and GELM (b) samples, with ΔEDT solution as function of the time of EDT change. The error bars indicate all the possible ΔEDT solutions and the color circles indicate the best-matching scenario.**





# 5 Discussion

## 5.1 Paleo-environmental $^3$He signal interpretation

All studied samples indicate the preservation of a $^3$He concentration consistent with temperatures that are colder than present-day EDT conditions at both the MBTP and the GELM sites (paleoIsoEDTs ~3-9°C and ~12-20°C lower than modern EDTs, respectively, Fig. 5). However, for both sites, the recorded $^3$He concentrations are apparently not concordant with simple time-EDT scenarios describing a plausible post-LGM mean temperature evolution in the European Alps (i.e., LGM mean temperature anomaly up to 15°C, Fig. 7). Even when allowing for a larger EDT difference between LGM and present-day (up to 40°C), modeled $^3$He concentrations remain significantly below the observed values at both sites. Such large EDT differences would furthermore not be supported by any mean temperature reconstructions for the European Alps since the LGM (e.g., Heiri et al., 2014a). Likewise, potential variation in seasonal temperature cannot contribute significantly to a larger pre-Holocene EDT anomaly. Indeed, global and regional paleoclimate studies rather suggest a larger seasonal temperature amplitude occurred before the Holocene (e.g., Davis et al., 2003; Buizert et al., 2018), which would have the effect of increasing the paleoEDTs instead (Fig. 9).

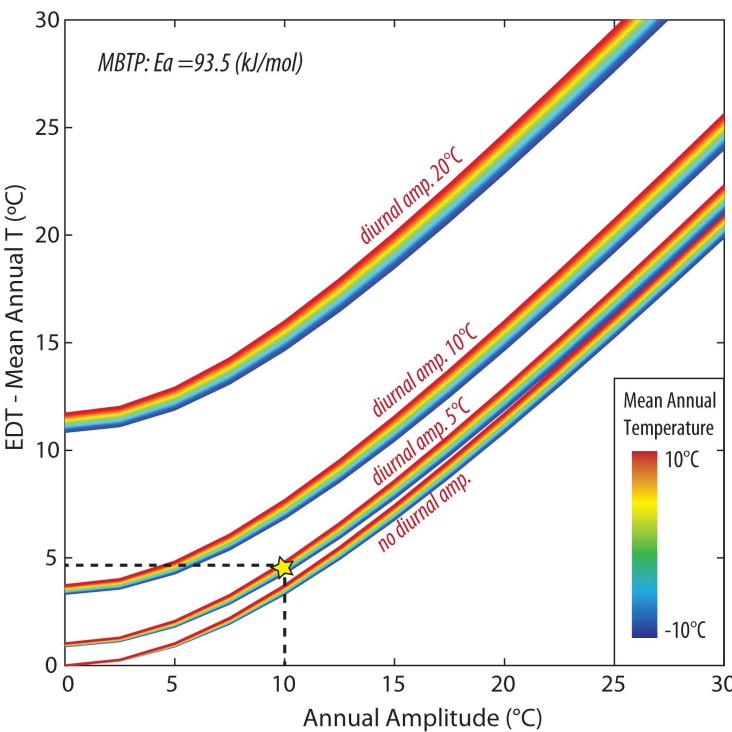

**Figure 9: Difference between EDT and mean annual temperature as a function of increasing seasonal temperature amplitude for different diurnal temperature amplitudes and assuming an E$_a$ of 93.5 kJ/mol (MBTP site, Holocene-calibrated diffusion kinetics). The yellow star indicates the conditions used to estimate the modern EDT at the sampling sites. Decreasing the annual/diurnal amplitude can yield to up to a ~5°C decrease in the modern EDT. Similar results are obtained when using Ea =98.5kj/mol (GELM site, Holocene-calibrated diffusion kinetics).**





We attribute the results of modeled $^3$He concentrations remaining significantly below the observed ones, despite significant

lowering of temperature prior the Holocene (e.g., Fig. 7), to the damping effect of the Holocene period (the last ~10 kyr), which is characterized by relatively stable mean temperature conditions similar to present-day. This hypothesis first appears to contradict our theoretical tests which indicate that $^3$He thermal signal inherited from past EDTs 10 to 15°C colder than today should *a priori* be (partly) preserved for 30-45 kyr under modern EDT conditions (Fig. 6). However, this time range relies on the assumption that bedrock surfaces were exposed for long enough to past colder conditions before the temperature change

occurred, in order to reach $^3$He steady-state concentrations (i.e., estimated exposure time around 20 kyr; Fig. 4). Along the MBTP and GELM profiles, bedrock surfaces have not been exposed for more than 5-8 kyr (MBTP) and 4-13 kyr (GELM) before the Late Glacial-Holocene transition, resulting in $^3$He accumulation up to 35-55% (MBTP) and 30-85% (GELM) of $^3$He steady-state concentrations when considering paleoIsoEDTs 10-15°C lower than present-day EDTs. In such case, $^3$He re-adjustment time estimates to modern EDTs are predicted to be reduced by ~90 to 80% for MBTP site and by >90 to 60% for

GELM site, implying we should recover the dominance of Holocene temperature conditions in the $^3$He signal from the sampled bedrock surfaces.

Our observed $^3$He concentrations can be reproduced by forward simulations with EDT change occurring on much more recent time scales (Fig. 8). For the MBTP site, a ΔEDT of 7 to 5°C within the last few thousand years to centuries predicts the observed natural $^3$He concentrations for two pre-Holocene samples (MBTP18-1 and 2; a ΔEDT of 12 to 8° is required for

MBTP18-11). Such a ΔEDT estimate, considering mean temperature fluctuations up to 2°C for the Holocene period (Davis et al., 2003), would also require variations in diurnal/annual temperature amplitudes to account for an additional 5°C ΔEDT. However, this would imply the lowering of both diurnal and annual temperature amplitudes to null before modern conditions (Fig. 9), which contradicts global and regional records that indicate an increased seasonality (and thus larger ΔEDT) in the early Holocene (Davis et al., 2003; Buizert et al., 2018) compared to the present-day. Furthermore, the forward simulations

discussed here (Fig. 8) used diffusion kinetics calibrated on Holocene samples. Therefore, allowing a significant EDT change over the last $10^2$-$10^3$ years is in contradiction with our calibration approach (cf. Sect. 3). If instead we use diffusion kinetics solely derived from laboratory experiments without Holocene calibration (Fig. 3, Table 2), an even larger recent ΔEDT is required to explain observed MBTP $^3$He concentrations (15 to 25°C or more than 30°C for changes over $10^2$ or $10^3$ yr, respectively, Fig. S2b). Such large ΔEDTs are significantly greater than expected EDT variations from changes in mean annual

temperatures and/or in annual/diurnal temperature amplitudes during the Holocene. Even greater ΔEDTs are needed to explain the observed GELM $^3$He concentrations using either diffusion kinetics approach: from 15 to more than 35°C (explaining GELM18-1 and 12 only, for EDT change over $10^2$ or $10^3$ yr, respectively) when using Holocene-calibrated diffusion kinetics (Fig. 8b), and to more than 40°C when using laboratory diffusion kinetics (no convergence found within the 40°C ΔEDT limit for pre-Holocene samples; Fig. S3b), which are in both cases clearly incompatible with plausible Holocene paleoclimatic

histories.

Finally, additional potential uncertainties in modern EDT estimates, used to define EDT of the recent and stable period in the step warming EDT scenarios (Sect. 4.3), cannot be ruled out. In particular, it is not known to what extent present-day conditions



(based on decadal direct air and ground temperature measurements; cf. Sect. 3.1) are representative over centennial to millennial time scales. Correcting for overestimated diurnal/annual temperature amplitudes and/or mean annual temperatures

would result in lower modern (i.e., recent) EDTs (Fig. 9). Assuming an overestimate of 50% in modern diurnal and annual temperature amplitudes, and up to 2°C overestimate in MARST based on recorded mean temperature fluctuations (Davis et al., 2003; Ghadiri et al., 2018, 2020) and applied corrections to MAAT (cf. Sect. 3.1), would lead to ~3.5°C lowering of modern EDTs for MBTP/GELM sites. Applying such an estimated correction to the recent period EDT potentially permits us to resolve observed $^3$He concentrations for two of the MBTP samples (MBTP18-1 and -2) with ΔEDT of 5 to 10°C for a change occurring

at ca.10 ka (i.e., LGM scenario; Fig. S2c). It is also worth noting that natural $^3$He MBTP concentrations for those samples can be reproduced with minor ΔEDTs (≤1.5°C) over recent timescales ($10^2$-$10^3$ yr). When using laboratory-derived diffusion kinetics without Holocene calibration, *a priori* more appropriate to explore recent EDT changes, only scenarios with more than -10°C ΔEDT within the last thousand years are accepted (Fig. S2d), inconsistent with paleoclimatic records over this recent time period. For GELM samples, correcting modern/recent EDT does not permit to reproduce the observed $^3$He

concentrations with plausible ΔEDTs for EDT change occurring at the Late Glacial-Holocene transition (ca. 10 ka; no solution), nor on more recent timescales (Fig. S3c-d).

## 5.2 Potential geological uncertainties

Several sources of geological uncertainties may affect the results obtained in this study. At first, our approach relies on the assumption that bedrock surfaces have experienced a simple exposure history along the time period recorded by $^{10}$Be

concentrations, without pre-exposure or episodic coverage. Depth profiles of $^{10}$Be measurements on glacially-polished bedrocks in the western Alps, with apparent exposure ages of 10-20 ka, indicate that an inherited $^{10}$Be concentration due to insufficient glacial erosion may persist and could lead to up to 9% age overestimates (Prud'Homme et al., 2020). Similarly, Wirsig et al. (2016b) suggested potential but limited pre-LGM (less than few ka overestimate) inheritance for some GELM samples. While previous bedrock surface exposure would also imply an inherited $^3$He concentration, the latter would be subject

to diffusion (partial or total) during glacier coverage (contrary to $^{10}$Be which would experience only minor radioactive decay over 10-100 ka timescales). Considering such a scenario (i.e., inheritance/complex exposure history) would hence result in lower $^3$He concentrations recorded by bedrock surfaces regardless of the temperature history experienced by the rock surface during the total $^{10}$Be exposure period (i.e., lower $^3$He/$^{10}$Be ratio; Balco et al., 2016). This scenario is also valid for post-LGM episodic coverage. Such effects are however expected to be minor considering the limited potential $^{10}$Be inheritance (<10%)

from pre-LGM exposure, as well as the unlikelihood of prolonged coverage of the relatively steep (i.e., no loose sediments/thick snow accumulation) and high (i.e., above tree line) sampled bedrock surfaces. Moreover, attempting to correct for these processes would result in opposite effects than what we observed for MBTP and GELM samples, with even lower paleoIsoEDT estimates and greater ΔEDTs required for warming EDT scenarios to recover observed natural $^3$He concentrations.



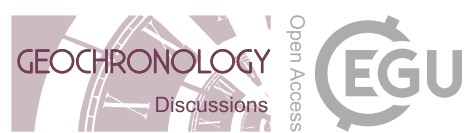

An additional source of uncertainty is postglacial erosion of sampled bedrock surfaces, assumed to be negligible in this study. Based on a combined approach exploiting cosmogenic $^{10}$Be and Optically Stimulated Luminescence (OSL) systems, Lehmann et al. (2020) suggested potential high postglacial erosion rates (above 3.5 mm/kyr) for low-elevation MBTP samples. Other regional estimates for crystalline bedrock commonly indicate Alpine postglacial erosion rates of 0.1 to 1 mm/kyr (Kelly et al., 2006; Dielforder and Hetzel, 2014; Wirsig et al., 2016b), in line with estimates from other studies (André, 2002; Balco, 2011).

Relatively low postglacial erosion rates are further supported along our study sites by the presence of still visible glacial striations (Wirsig et al., 2016b). Applying an erosion correction (0.1 to 1 mm/kyr) will only moderately affect apparent $^{10}$Be exposure ages (<1 ka change), and would result in lower predicted $^3$He concentrations compared to our observed ones.

   In summary, geological uncertainties related to exposure history and postglacial surface erosion are generally small and overall do not resolve the significant discrepancy between the natural $^3$He signal recorded in pre-Holocene MBTP and GELM samples

and modelled $^3$He concentrations from expected EDT histories.

   On the other hand, some of the observed differences may relate to uncertainties regarding the $^3$He production rate ($P_{3He}$) in quartz. Directly estimating $P_{3He}$ in quartz from geological calibration sites is challenging, as $^3$He diffuses from quartz at Earth surface temperatures over $10^2$-$10^4$ yr time scale. Alternative approaches using artificial targets (e.g., Vermeesch et al., 2009) or scaling $P_{3He}$ measured in other retentive minerals (i.e., feldspar; Masarik and Reedy, 1996) have hence been used. While in

this study we adopted the Stone (2000)-scaled $P_{3He}$ from Vermeesch et al. (2009; i.e., 116 at.g$^{-1}$.yr$^{-1}$), a 10% higher $^3$He production rate has also been proposed (e.g., Ackert et al., 2011). Applying an increased $P_{3He}$ (Stone-scaled $P_{3He}$= 128 at.g$^{-1}$.yr$^{-1}$) in general leads to smaller ΔEDTs in order to match the measured $^3$He concentrations, as well as an older range of possible times for the EDT change. For MBTP site, however, we could not reproduce $^3$He concentrations for an EDT change at 10 ka (except for MBTP18-1; Fig. S2e, Holocene-calibrated diffusion kinetics). Likewise, for more recent changes ($10^2$-$10^3$

yr; laboratory-derived diffusion kinetics without Holocene calibration; Fig. S2f), the resulting ΔEDTs (10 to 25°C) are still not compatible with plausible Holocene temperature conditions. Similar results were obtained for the GELM Late Glacial samples when adopting a 10% increase in $P_{3He}$ (Fig. S3e, f).

   In addition to a higher cosmogenic $^3$He production rate, another possibility that we have not accounted for is non-cosmogenic sources of $^3$He, specifically nucleogenic $^3$He produced by (n,α) reactions with $^6$Li. Unaccounted for nucleogenic $^3$He would

result in lower true cosmogenic $^3$He concentrations, which would have the effect of reducing the ΔEDTs at our sample sites toward more realistic values. However, we think it is unlikely that there is significant nucleogenic $^3$He in our samples for several reasons. Based on the diffusion kinetics of $^3$He in quartz, we anticipate that any nucleogenic $^3$He produced in our samples over geologic timescales will be diffusively lost before the quartz is exhumed at near-surface temperatures. Furthermore, the production rate of nucleogenic $^3$He is low compared to the cosmogenic production rate of $^3$He. We do not

have major and trace element data for the MBTP and GELM samples in order to calculate the nucleogenic $^3$He production rate directly. However, we can say that a rough maximum estimate for the production rate of nucleogenic $^3$He in the GELM samples is ~1 at/g/yr, which is based on a maximum Li concentration of 70 ppm for the Aare granite (Schaltegger and Krähenbühl, 1990) and the production rate estimate of Farley et al. (2006) for an 'average' granite. This is 0.3% of the local, scaled



production rate of cosmogenic $^3$He for sample GELM18-9, which has the lowest cosmogenic $^3$He production rate of all of our samples. The combined low retentivity and small production rate of nucleogenic $^3$He indicate that this does not contribute significantly to our measured $^3$He concentrations, and is therefore unlikely to affect our modeled ΔEDTs in any significant way.

### 5.3 $^3$He diffusion kinetics and $^3$He thermal signal

Cosmogenic $^3$He paleothermometry is still in its early-stage of development for application to Quaternary geology (Tremblay
et al., 2014a, b; 2018). At present, there are nontrivial uncertainties related to the interpretation of $^3$He diffusion kinetics in quartz, specifically regarding how to extrapolate diffusion kinetics data obtained in laboratory experiments down to Earth surface temperatures in order to interpret natural cosmogenic $^3$He concentrations.

Noble gas diffusion in minerals is generally assumed to have an Arrhenius-type dependence on temperature, where diffusivity increases exponentially with temperature, and inversely with the diffusion domain size (e.g., Baxter, 2010 and references
therein). Interestingly, theoretical studies investigating the fundamentals of $^3$He diffusion in quartz predict considerably lower $E_a$ (and much higher diffusivity) than expected when considering a perfect quartz crystal (~20 to 50 kJ/mol; Kalashnikov et al., 2003; Lin et al., 2016; Domingos et al., 2020; Liu et al; 2021), the latter suggesting that no $^3$He should be retained over geological timescales at Earth surface temperatures. These results are, however, in contradiction with common observations of $^3$He retention in natural rock surfaces (e.g., Brook et al., 1993; Brook and Kurz, 1993; Tremblay et al., 2018) and with
typical $E_a$ values empirically determined from laboratory-degassing experiments (between 70 to 100 kJ/mol; Shuster and Farley, 2005; Tremblay et al., 2014b). Furthermore, previous $^3$He-degassing experiments conducted on quartz from various origins indicate a large variability in diffusion kinetics (i.e., $E_a$ and $D_0$) and diffusion behavior, wherein some quartz samples exhibit complex $^3$He diffusion behavior while others exhibit a simple, linear Arrhenius dependence (Tremblay et al, 2014b). Both the observed variability and the discrepancy with theoretical predictions suggest that $^3$He diffusion in natural quartz is
largely governed by sample-specific crystal defects (e.g., structural defects, radiation damages; Domingos et al., 2020), advocating for the use of sample-specific diffusion kinetics (Tremblay et al., 2014b). Complex, non-linear diffusion behavior has been previously observed for argon diffusion in feldspar (e.g., Berger and York, 1981; Harrison and McDougall, 1982) that is analogous to the complex $^3$He diffusion behavior observed in some quartz samples. Lovera et al. (1989; 1991) proposed a multi-diffusion domain (MDD) model to account for complex argon diffusion behavior which describes the simultaneous
diffusion of discrete, non-interacting intracrystalline sub-domains (e.g., sub-grain fragments) characterized by different effective diffusion lengthscales. Tremblay et al. (2014b) applied the MDD model framework to $^3$He diffusion in quartz for samples that exhibited complex Arrhenius behavior, and we have adopted the same approach here.

However, it remains an open question as to whether MDD-type models are applicable to quartz $^3$He paleothermometry. In Antarctica (Pensacola Mountains), both a single-diffusion domain model using diffusion kinetics from Shuster and Farley
(2005) and a two-domain model using kinetics from four local erratics could successfully explain the $^3$He signal observed in a series of Holocene samples (Tremblay et al., 2014a; Balco et al., 2016), with a similar predicted $^3$He concentration evolution



between the two approaches over this timescale (Balco et al., 2016). However, each approach could only partially explain the ³He signal recorded in samples with older ¹⁰Be exposure ages, with complex exposure history and/or significant inter-sample variability in diffusion kinetics (e.g., different quartz sources for the sandstone lithology) likely acting as compounding factors

(Balco et al., 2016). Additional quartz ³He analyses using a MDD model and sample-specific diffusion kinetics were recently conducted on moraine boulders from the Gesso Valley in the Italian Alps with LGM to Late Glacial chronologies (Tremblay et al., 2018). PaleoIsoEDTs within the range of their respective modern EDTs were obtained for two out of five samples, with no clear trend between paleoIsoEDTs and boulder (¹⁰Be) exposure ages/relative moraine age, in addition to significant intra-moraine variability. Tremblay et al. (2018) highlighted multiple sources of potential uncertainties related to local shading

effects (i.e., vegetation, snow cover, topography), grain-size scaling, and complex boulder exposure histories which could have contributed to the observed ³He signal inconsistencies.

In this study, bedrock-surface samples were purposefully collected along high-elevation valley profiles progressively deglaciated between the LGM and Holocene, with the aim to limit the potential for complex exposure (cf Sect. 5.2). Diffusion kinetics parameters were measured on one representative sample per profile (MBTP18-9 and GELM18-1). Although inter-

sample diffusion kinetics variability cannot be excluded, the apparent homogeneous igneous lithology along each profile supports the representativeness of our chosen sample per profile for diffusion kinetics experiments. Based on this first-order assumption, we noted different ³He diffusion trends between MBTP and GELM representative samples. MBTP quartz exhibits a nearly simple (i.e., linear; Fig. 3a) Arrhenius diffusion behavior, and measured ³He concentrations recorded along the MBTP profile can potentially be interpreted as at quasi-equilibrium with respect to modern EDTs (despite a slight trend towards colder

signal) when considering the potential sources of uncertainty (e.g., Holocene EDT, $P_{3He}$ etc., Sect. 5.1 and 5.2; based on the Holocene-calibrated diffusion kinetics). On the contrary, GELM quartz is characterized by complex ³He diffusion behavior (Fig. 3b), and bedrock surfaces record a ³He thermal signal that is apparently well colder than their modern EDTs when using diffusion kinetics derived from a MDD framework (calibrated on Holocene samples). This apparent divergence cannot be resolved within the multiple sources of geologic uncertainties, nor it can be explained by plausible fluctuations in thermal

variables (i.e., mean annual temperature and diurnal/annual amplitudes) during the Late Glacial and Holocene time periods. One possible interpretation of these results would be that the MDD model we applied to the GELM samples does not accurately represent ³He diffusion in quartz that occurred during exposure time. This could be because the MDD model does not adequately represent the physical process of ³He diffusion in quartz. From a mineralogical perspective, it is indeed unclear if potential processes involved in the formation of sub(-grain) domains (e.g., cooling, alteration, deformation) are consistent with

the assumed conditions of the MDD model, i.e., disconnected sub-domains with fixed volumes, Fickian and isotropic diffusion, and zero concentration boundary conditions (e.g., Lovera et al., 1991; Baxter et al., 2010). While MDD models have been successfully applied in a number of thermochronology applications (Reiners et al., 2005 and references therein), deformation processes may also lead to interconnected sub-grain microstructures (e.g., Reddy et al., 1999), in which case the MDD model may be inappropriate for obtaining accurate thermal constraints, as already acknowledged in the literature (e.g., Lovera et al.,



2002; Harrison and Lovera, 2013). On another hand, alternative diffusion models involving multi-path diffusion (e.g., Lee, 1995) are also suffering from substantial theoretical and experimental gaps (Harrison and Lovera, 2013; Baxter et al., 2010). Alternatively, we cannot rule out that a MDD model for quartz [3]He paleothermometry (Tremblay et al., 2014b) is applicable on both MBTP and GELM quartz, but that the diffusion kinetics and/or the predictive [3]He diffusion model over Earth surface temperatures are inaccurately constrained. The MDD models we implemented do not provide unique solutions to our

laboratory-measured diffusion kinetics, which we then extrapolate down to Earth surface temperatures (<30°C). This is illustrated by the significant difference between modern EDTs and estimated paleoIsoEDTs observed for Holocene samples (both MBTP and GELM sites) when using laboratory-derived diffusion kinetics without Holocene calibration (Fig. S4), which therefore supports the additional Holocene calibration step applied in this study. However, our chosen approach still remains relatively crude considering all possible uncertainties related to samples thermal and exposure history (Sect. 5.2 and 5.3). As

a consequence, we may consider that the apparently colder signals recorded by [3]He analyses along both profiles (although less pronounced in MBTP) are real, but that these cannot currently be well quantified in terms of surface paleoEDTs. We also compiled all quartz [3]He paleoIsoEDTs available in the European Alps (Tremblay et al., 2018; Guralnik et al., 2018; this study; Fig. 10). Interestingly, while this compilation reveals no apparent relationship with [10]Be exposure age (from LGM to Holocene; Fig. 10a), we observe an apparent negative correlation between samples paleoIsoEDT and elevation (Fig. 10b). Furthermore,

while samples at low to moderate elevations have paleoIsoEDTs that are relatively consistent with their estimated modern EDTs along an apparent linear lapse rate (around -0.5°C/100m lapse rate), paleoIsoEDTs recorded in rock surfaces above ~2200 m a.s.l. clearly depart from modern EDTs/lapse rate trend with significantly "colder" [3]He signals. Although the compiled Alpine dataset is still limited, such an observed distribution raises the question of the influence of rock-surface elevation on [3]He signal records. One hypothesis is that [3]He release from quartz minerals over geological timescales is less effective than

predicted by current [3]He diffusion models for the colder temperatures ranges inherent to high-elevation settings. Alternatively, we can suggest that the recorded "colder" [3]He signals in high-elevation samples may reflect recent changes in Alpine permafrost ground conditions. Indeed, bedrock-surface samples above ~2200 m are located close to or in the lower range of sporadic to discontinuous permafrost distribution in the present-day Alps (Magnin et al., 2015a; Boeckli et al., 2012b). Recent warming after the Little Ice Age is expected to have led to permafrost degradation and restriction of its spatial distribution

towards higher elevations (Magnin et al., 2015a, 2017). We hence cannot exclude that those high-elevation bedrock surfaces may have experienced permanent permafrost conditions until recently (i.e., last tens to hundreds of years), where the past MARSTs were thus lower (sub-zero range) than modern MARST estimates scaled on mean annual air temperature (Table 1; Sect. 3.1.). In that case, the recent change in climate conditions over the last decades to centuries would have resulted in both mean annual temperature increases and amplification of annual and diurnal temperature oscillations at the sampling sites

greater than those constrained from air temperature records (Etzelmüller et al., 2020) due to the transition from a permafrost to non-permafrost zone. To test these hypotheses (i.e., [3]He diffusion inaccurately constrained at Earth surface temperature and recent permafrost degradation effects) would require further quartz [3]He measurements at high-elevations and in other Alpine/cold regions.





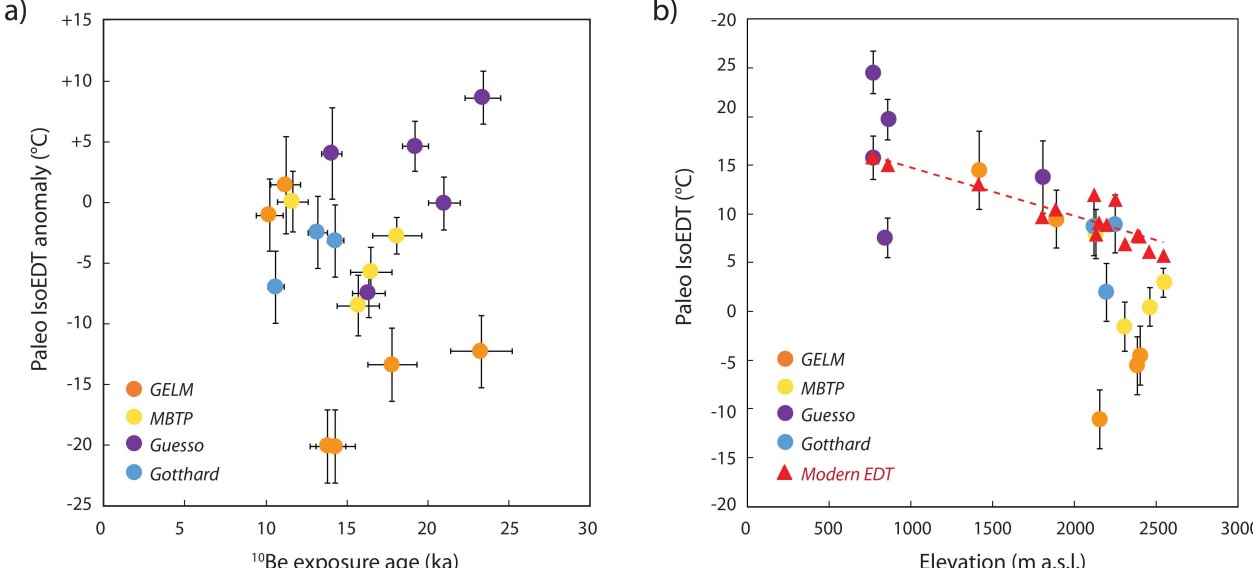

**Figure 10: (a) Relationship between ³He paleoIsoEDT anomaly and ¹⁰Be exposure age from available data from moraine boulders and glacially-scoured bedrock surfaces in the Alps (results from this study, Tremblay et al., 2018 (Guesso) and Guralnik et al., 2018 (Gotthard)). (b) Relationship between ³He paleoIsoEDT and elevation for same dataset as in (a).**

## 6 Conclusion

Paleoglacier fluctuations in alpine settings lack direct constraints of associated past temperature and/or precipitation
conditions, essential to improve our understanding of the response of glaciers and (para)glacial processes to past and future
climate forcing changes. In this study, we applied quartz ³He cosmogenic paleothermometry to derive *in situ* paleo-temperature
(EDT) estimates along two deglaciation sequences gradually exposed from the Last Glacial Maximum to the Holocene in the
western/northern European Alps (Mont Blanc and Aar massifs, MBTP and GELM respectively). Investigation of quartz ³He
diffusion kinetics indicates a clear difference between the two study sites, with quasi-linear *vs.* complex diffusion behaviors.
Based on the assumption that same diffusion kinetics parameters apply to all samples at each site, forward numerical
simulations of ³He production and diffusion suggest that no thermal signal from the Late Glacial period should be preserved
in investigated rock surfaces with brief exposure durations (several kyr) before the transition to relatively stable Holocene
climatic conditions like present-day. However, all our rock-surface samples exposed prior to Holocene indicate an apparent
³He thermal signal significantly colder than present-day conditions. Our recorded ³He signals cannot be explained by realistic
post-LGM mean annual temperature evolution in the European Alps (as recorded by other paleoclimatic proxies), neither by
changes in annual and/or diurnal temperature oscillations at the study sites.

When accounting for potential uncertainties related to Holocene thermal conditions and quartz ³He production rate, the ³He
signals (ΔEDT) recorded along the MBTP site can potentially be interpreted to be close to equilibrium with present-
day/Holocene conditions, with minor change in mean annual temperature or diurnal/annual temperature oscillations. However,

$^3$He derived paleo-EDTs along the GELM site remain distinctively colder than present-day conditions. One hypothesis is that the multi-diffusion domain models applied to characterize the observed complex diffusion behavior in the GELM quartz does not accurately quantify quartz $^3$He diffusion in the samples of this site throughout their exposure histories. Alternatively, if the general colder trend recorded along both profiles is possible, the assumed quartz $^3$He diffusion kinetics and diffusion models may inaccurately extrapolate to Earth surface temperatures, precluding quantitative EDT constraints from the observed $^3$He

abundances in these samples. Finally, considering the high elevations of the investigated rock-surface samples (> 2000 m), it is also possible that our $^3$He thermal signals result from much more recent changes in Alpine permafrost ground conditions during the past decades/centuries. While data presented in this study demonstrate the promising use of $^3$He cosmogenic paleothermometry to quantify past environmental changes, additional $^3$He analyses in high-alpine/cold settings would be necessary to clarify to which phenomena is the $^3$He thermal signal most responsive, i.e., between Late-Pleistocene ambient

temperature variations and recent changes in permafrost distribution.

**Code availability**

The source codes (with examples of input dataset) used to determine (1) $^3$He diffusion kinetics from step-heating experiment applying a MDD model framework (example diffusion data from MBTP18-9) and to (2) conduct forward simulation of $^3$He production/diffusion along given time-EDT scenario (simplified LGM scenario for MBTP18-1 as example, Fig. 7) are

available on Zenodo at https://doi.org/10.5281/zenodo.5808021 (Tremblay, 2021).

**Data availability**

No additional data are used in this paper that are not supplied in the Supplement.

**Supplement link**

The supplement related to this article is available online

**Author contributions**

NG and PGV designed the study. NG led fieldwork campaigns, with support of BG, and prepared samples for laboratory analysis. NG, GB, MMT and DLS conducted the measurements. NG performed the numerical experiments using the model developed by MMT and GB. NG led the manuscript preparation, with contributions from all co-authors to the analysis and interpretation of the data, manuscript writing and review.



**Competing interests**

Some authors are members of the editorial board of Geochronology. The peer-review process was guided by an independent
editor. The authors declare no other conflict of interest.

**Acknowledgments**

This work was supported by the Swiss National Science Foundation (Grant PP00P2_170559), with measurements at BGC in
part supported by the Ann and Gordon Getty Foundation. PGV acknowledges support from the French ANR-PIA program
(ANR-18-MPGA-0006). MMT acknowledges support from the U.S. National Science Foundation (OPP-1935945) and the
AAAS Marion Milligan Mason Fund. We thank Benjamin and Fred Lehmann, and Guilhem A. Douillet for their help in the
field to collect samples.

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
