# Peer review of "Cosmogenic 3He paleothermometry on post-LGM glacial bedrock within the central European Alps"

_Geochronology, 2022_

## Referee Comment (RC2)

*"Cosmogenic $^3$He paleothermometry on post-LGM glacial bedrock within the central European Alps"*

by N. Gribenski et al.

Comments by reviewer Samuel Niedermann

*General comment*

The authors have measured cosmogenic $^3$He in quartz from two Alpine altitude profiles, with the aim to study paleo-temperature conditions in these areas that were gradually deglaciated after the LGM. They did the diffusion experiments that are required for a correct interpretation of the data and applied a lot of care in devising their experiments and model calculations, but obtained results that are difficult to interpret because temperatures recorded by cosmogenic $^3$He are generally lower than expected based on the models. They discuss possible explanations for the discrepancy, but cannot give a final answer. Nevertheless, I think this is an important paper as it shows the difficulties (but also the potentials) involved in the rather new cosmogenic $^3$He paleothermometry method and may thus prevent other scientists from misinterpreting their own data. With regard to the methods used, there is one thing that was not optimal in my opinion: Obviously, the helium analyses were confined to $^3$He (at least nothing else is reported). Measuring $^4$He also would have revealed any possible contributions of nucleogenic $^3$He, which may lead to an overestimation of cosmogenic $^3$He as argued in more detail below.

For the most part, the paper is written clearly and concisely. The English is generally fine, but suffers from quite a lot of small minor defects such as missing articles etc., which is a bit annoying given the fact that at least three co-authors are native English speakers. I have tried to point these out in the manuscript. Anyway, I recommend this manuscript for publication in *Geochronology* after minor revision has taken account of the specific and technical comments given hereafter.

*Specific comments:*

- In lines 172-182, the authors briefly describe the methods used for $^3$He determination. Surprisingly for me, they don't mention anything about $^4$He, and because there are no $^4$He or $^3$He/$^4$He data in the tables either, I assume they have not even checked for $^4$He concentrations (otherwise those data should be given). This is a pity in the context of the discussion that many samples seem to contain more cosmogenic $^3$He than expected (see below).

- Line 177: Why were blanks (only) measured at room temperature? Did you ever check whether they remain the same at higher temperatures up to the level used in the experiments? Otherwise that assumption seems a bit optimistic.

- In lines 523-537 the authors discuss a possible contribution of nucleogenic $^3$He in the samples that has not been accounted for. They conclude that it "does not contribute significantly", but their arguments are not very convincing because they have (obviously) not measured $^4$He. It is known that purely radiogenic/nucleogenic He is characterized by typical $^3$He/$^4$He ratios of ~$(2\pm2)\times10^{-8}$. Thus, if the measured $^3$He/$^4$He ratios in their

samples were on the order of $10^{-6}$ for example, the nucleogenic $^3$He would be in the percent range at most and could safely be neglected. In my lab, I have measured several samples from similar settings (Mont Blanc area as well; work in progress) using two extraction steps of 600°C and 1000°C. While the 1000°C steps showed ratios in the $10^{-8}$ range as expected for radiogenic/nucleogenic He, the ratios in the 600°C steps varied from ~$6\times10^{-8}$ to several times $10^{-6}$, implying that the nucleogenic $^3$He contribution is not always negligible even in the low temperature step. Since the authors used 800°C as the first heating step, this is even more relevant for their samples. Their arguments for negligible nucleogenic $^3$He are not very powerful. Assuming the same diffusion characteristics for nucleogenic and cosmogenic $^3$He is probably not valid, since nucleogenic He is produced at different places in the minerals (namely, where Li is found, which is probably concentrated in mineral or fluid inclusions rather than in the quartz crystal itself). The fact that nucleogenic He is still degassed above 600°C while cosmogenic He is not shows on its own that the diffusion characteristics are different. Also, comparing the production rates of cosmogenic and nucleogenic He is not meaningful at all, since nucleogenic He has been produced over many millions of years, compared to the ~10 ka production of cosmogenic He. I don't say that the overabundance of cosmogenic $^3$He in the authors' samples is indeed due to nucleogenic $^3$He, but because of the lack of $^4$He data it is difficult to rule it out.

*Technical comments:* (numbers refer to line numbers in the manuscript)

5-10   Please give zip codes of European cities before the city name (without comma), as customary here.

19     Should be "... in quartz from the Mont Blanc site and and complex Arrhenius behavior in quartz from the Aar site...", as "behavior observed from the Aar site" is an odd wording.

31     Tremblay et al. 2014a or 2014b?

57     It is odd to say "glaciers retreated quickly behind the Little Ice Age moraines" when considering a time long before these moraines were there! Change to something like "behind the position were today the Little Ice Age moraines are located".

64     I assume this should rather read "High-resolution $\delta^{18}$O in Alpine speleothems …"

70     "which resulted in … North Atlantic atmosphere patterns"??? Do you mean in variable atmosphere patterns? Then you should repeat that word (it's in singular in the first part, so it doesn't seem to belong to the plural term in the second one).

77     There is no Bartlein et al. 2014 in the reference list (only B. et al. 2010).

79     Please explain ELA for those readers who are not familiar with that term.

132   "from the exact same locations previously collected" seems to imply you sampled surfaces from which some rock had already been knocked off before, i.e your surfaces would have been covered until a few years ago. I'm sure that's not what you did, but please clarify it!

Fig. 2:   On the y axis labels, remove dot after m because the symbol m for meter is never written with a dot. In the caption, it is confusing to write (a-c), (b-d) which seems to mean a to c and b to d, while obviously a and c, b and d is meant. So change to (a, c), (b, d).

Table 1: Lehmann et al. 2019 and Borchers et al. 2016 are not found in the reference list (but L. et al. 2020, B. et al. 2015). Give details about the method to estimate Mean Annual Rock Surface Temperatures. Please explain EDT or, at least, refer to the text section where it is explained.

184   Tremblay et al. (2021): Only Tremblay (2021) in reference list.

191   Is this the spallogenic or total $^{10}$Be production rate?

192   Borchers et al. 2016 see above (Table 2)

243   What is "0.5 increment"? Do you mean increments of 0.5 kJ/mol? If so, you must give the unit!

244   Baxter et al. 2010 is not in reference list (only Baxter 2010).

Fig. 3:  I can't see any gray lines, just gray areas between red and blue lines! This whole figure is very confusing and difficult to decipher, and the explanations in the caption do not help a lot. I ask the authors to think about how this figure could be improved, e.g. by showing several panels per location, and how it could be explained better.

284   Related to the above, I don't understand how the parameters $E_a$ and $\ln(D_0/a^2)$ can be obtained from Fig. 3.

Table 2:  4th column from left, MBTP d3: Range 8.67 to 10, is that 8.67 to 10.00? If so, don't omit the .00, because otherwise it looks like this value was much less precisely determined than the other ones.

Fig. 4:  At such a limited range of values (~2000-20,000), I don't see why a logarithmic scale is used for the y axis.

341-342  I assume what you mean is "… from the highest sample (…) partially overlaps that from MBTP18-9 within uncertainty."

Fig. 5:  What is the difference between the solid and dashed green lines in panels e and f?

374   Instead of "experiments", which may imply something physical, better write "model runs" or similar.

487   Prud'Homme or Prud'homme as spelled in reference list?

514   Feldspar is not a mineral that is retentive for $^3$He; actually it is even less retentive than quartz! I don't think Masarik and Reedy have based their numbers on feldspar.

533   Farley et al. 2006 is not in reference list.

547   Liu et al. 2021 is not in reference list.

557   Berger and York 1981 is not in reference list.

596, 601  Baxter et al. 2010 is not in reference list (only Baxter 2010).

Data supplementary:

Fig. S1  Please use consistent sample labels throughout the manuscript. Here, GELM18-1 and GELM18-6 is written in the legend but GELM18-01, -06 in the figure caption.

Tab. S1, S2:  The uncertainties given for $^3$He are unrealistically small, in some cases <1 permil! This is obviously nonsense, no mass spectrometer is able to measure as precisely as that. I assume the authors only considered the statistical error of the measurement, disregarding other error sources such as mass spectrometer sensitivity or linearity. Even if the systematic errors (such as the uncertainty of the He concentration

in the standard that is used for sensitivity determination) are not taken into account, which makes sense in such a step-heating experiment (but should be noted), short-term variations of the sensitivity or deviatons from the calibrated or assumed linearity of the mass spectrometer signal will be in the percent rather than the permil range.

Tab. S3: Similar to the $^3$He uncertainties in Tab. S1 and S2, here the $^{10}$Be uncertainties in the permil range are unrealistic. Again, it looks like only the $^{10}$Be/$^9$Be measurement error was taken into account, without considering the uncertainty of the standard or carrier for example.

Footnote 2 should refer to Table S4 rather than Table S2.

Tab. S3, S4: In sharp discrepancy from the former two tables, $^3$He errors are very high (on the order of 10%) here. While this may be realistic, the reason for the much higher uncertainties should be explained even when those in Tables S1 and S2 have been adjusted to a more realistic level.

The $^3$He concentrations and their uncertainties are given with unreasonable precision, i.e 6 significant digits for the errors. It is inappropriate (and confusing) to give more than two significant digits for an uncertainty, because uncertainties are not precise numbers but just represent probabilities. Therefore, values such as 520,929 should be rounded to 520,000, and of course the corresponding values should always be given with the same precision as the uncertainties. The first entry in Table S3 would thus be 6,400,000 ± 520,000, or better readable (and easier to round in Excel, e.g.) 6.40 ± 0.52 in units of $10^6$ at/g.

I wonder how the average values from the three replicate measurements were calculated. For three measurements agreeing within error limits, as seems to be the case looking at Table S4, an error-weighted mean could be calculated. Obviously, the authors calculated an unweighted average instead and, very strangely, used the average of the errors as the error of the average. I don't think this is a statistically correct way.

[revised manuscript text omitted]

---

## Author Comment (AC2)

Comment on gchron-2022-1

Gribenski et al. (2022)
Geochronology Review

**Referee #2 (Samuel Niedermann)**

[Figure]

Figure RN1: Amounts of $^3$He and $^4$He released in the 800ºC heating steps in samples from the MBTP (left) and the GELM (right) transects

[Figure]

Figure RN2: Solutions of time-EDT scenario explaining our observed $^3$He concentrations for the MBTP samples obtained using $^3$He errors either derived from the average error (left panel, as done in our study) or from the standard deviation (right panel)

---

## Author Response (AR1)

**Point-by-point response to the reviews received for the manuscript GCHRON-2022-1 "Cosmogenic $^3$He paleothermometry on post-LGM glacial bedrock within the central European Alps" by Gribenski et al.**

*Black font: original reviews*
*Green font: detailed responses with the description of all the relevant changes made in the manuscript and the supplement material*

**Anonymous Referee #1**

Gribenski et al. (2022)
Geochronology Review

In this manuscript, Gribenski et al. detail the methodology of using diffusion kinetics and cosmogenic $^3$He measured in quartz crystals to quantify paleo-surface temperatures for two glacial valleys in the European Alps. Description of laboratory methods and diffusion systematics present an argument for how they can be theoretically used to decipher surface temperatures in paleo-environments. However, results from all samples indicate regional temperatures in the pre-Holocene colder than records from other regional proxy studies. Potential influences on this temperature discrepancy could come from issues with helium diffusion kinetics within the measured quartz crystals or geologic/paleoclimatic uncertainty of the sampled glacial surfaces.

This is manuscript is very thorough and comprehensive in describing the theory and concept of $^3$He diffusion and use as a paleotemperature proxy, testing of results through modeling experiments, while acknowledging areas of misfit and uncertainty. The study presents a novel approach to determining a scientifically relevant problem valuable to the paleoclimate community – what are past surface temperatures? Detailed description of the methods and results allow for reproducibility and illustrate consideration by the authors of multiple variables/influences on the results. While the data presented do not result in a fully-realized and issue-free method of determining EDT, they represent an important step forward in achieving that goal. All interpretations and conclusions are supported by the data. Furthermore, by acknowledging areas of uncertainty and external influences (e.g., permafrost) the authors provide the groundwork for future studies to build off these results.

We thank the reviewer for their overall positive reception of the study and the constructive comments.

While the manuscript is written logically, the overwhelming number of analyses left me lost from time to time. In particular, in the results section, description of the modeling experiments had me revisiting earlier sections to recall why each experiment was being tested. Adding a synoptic sentence or two for each results section will assist the reader in appreciating the robust quality of the experiments within this study.

In order to facilitate the reading of the results section, and in line with another comment from Reviewer 3 (4. Result section, General comment) we will add an introductory paragraph summarizing the different results reported in each following subsections: *"First, we examine the characteristics of the $^3$He diffusion kinetics parameters we modeled for our quartz samples and explore their sensitivity of the $^3$He signal in those samples to Earth surface EDTs. We then present forward model results for the evolution of the cosmogenic $^3$He concentrations recorded along each deglaciation profile for two different sets of thermal histories. The first set of thermal histories we investigate assumes a constant EDT since the exposure of the sampled rock surfaces following ice retreat. We then investigate a set of more climatologically-*

*interesting thermal histories, wherein a change in EDT occurs at some point during the exposure time of each sample"* (line 319-324).

We will also revise the section headings to delineate which sections are about methods, and add a brief synoptic summary at the beginning of the methods section (section 3) about different methods used. We think this will help clarify the manuscript significantly and make the number of different analyses more digestible.

In addition, we will implement the manuscript following the more specific comments/edits suggested below.

Below are a few minor comments/edits for the manuscript which should otherwise be accepted for publication:

Line 123 – Mention the two interpretations for glacial trimlines: ice-surface vs. thermal boundary. However, this is not mentioned again during the discussion of results, nor is one interpretation suggested over the other. With potential nuclide inheritance discussed later, could coverage by non-erosive, cold-based ice influence the EDT values for a particular site based on the thermal boundary interpretation?

The samples have been all collected below the trimline, implying that during the LGM, they were covered by ice. We do not hypothesize on the ice thickness and thermal state of the ice covering the sampled rock surfaces at this time (i.e., during the LGM) as this is out of the scope of this study.

We are discussing the implications of episodic ice coverage or inheritance due to insufficient erosion (e.g., due to cold-based ice) on the $^3$He/$^{10}$Be ratio and paleoIsoEDT interpretations in section 5.2 ("Potential geological uncertainties"; that will become section 5.2.2 "Interpretation of cosmogenic nuclide measurements" in the revised manuscript). To make a clearer parallel with possible cold-base ice coverage, and to further illustrate the effect of ice coverage on $^3$He evolution we will:
- refer to possible ice coverage by non-erosive cold-based ice: "*First, our approach relies on the assumption that [...], without pre-exposure or episodic coverage (i.e., non-erosive cold-based ice)*" (line 533-536).
- specify that $^3$He will continue to diffuse even at subzero temperatures as expected under ice cover: "*While previous bedrock surface exposure would also imply an inherited $^3$He concentration, the latter would be subject to diffusion (partial or total) during glacier coverage, even at subzero temperatures and EDTs (Fig. S4)*" (line 541-542).
- add a figure in the data supplementary (Figure S4) illustrating the diffusion of $^3$He using diffusion kinetics parameters determined for both study sites, for isoEDTs as low as low as -30°C.

Line 170 – Briefly describe the purpose of the tantalum packets for those unfamiliar with the lab methodology.

The use of tantalum packets is not mandatory. What is important is that the samples are packed in some sort of metal packet because metal couples well with diode laser, and because the emissivity of the chosen metal can be calibrated so that we know the temperature that we are heating it to via pyrometer measurements.
In our experiments we were actually using two type of metals: tantalum for bulk degassing experiments and platinum-iridium (PtIr) for stepwise-heating experiments.

We will add this information together with the temperature control systems: "*For bulk degassing measurements (Tables S3-S4), the samples were packaged into tantalum packets and heated in two, 15-minute long heating steps at 800 and 1100 °C with a diode laser, with temperature of the tantalum packet measured by pyrometry.*" (line 200-202); "*For stepwise-heating experiments on proton-irradiated grains, the selected grains were placed in contact with the tip of a bare wire K-type thermocouple inside small platinum-iridium (PtIr) packets. The PtIr packets were heated with a diode laser in a feedback control loop with the thermocouple.*" (line 205-207)

Line 176 – Is there a set increment of temperature increase? If so, it might be easier for the reader to interpret those changes rather than the number of increases within a time range.

The increment of temperature change between steps varies; we will simply say this now to avoid confusion about how it is written and refer the reader to the data supplementary table where the holding temperatures are given.

Line 190 – How sensitive are the results for 10Be ages and EDT calculations if a different scaling scheme is used (e.g., time-dependent LSDn)?

Exposure ages calculated using the time-dependent LSDn scaling scheme can be up to 7 % different from those calculated with the time-independent St model, as used in this study.
For such a difference, we estimate that the influence on the EDT calculation (e.g. paleo IsoEDT) remains negligible (<1°C) and within the error margin.

Line 196 – From where are the initial EDT temperatures initially derived and how? A little clarification is helpful.

A full section (3.2; which will become section 3.5 "Effective diffusion Temperature estimates" in the revised manuscript) is dedicated to the description of how the initial (modern) EDT at each sampling site is derived, with an accent on temperature variables used for this calculation. In addition, to further clarify how past EDTs in the case of forward 3He modelling for varying thermal history) were estimated, we will add at the end of the section: "*Past colder EDTs input in forward [3]He modelling for varying thermal history are based on temperature anomalies from modern EDTs.*" (line 302-303)

We will also reorganize and edit the Methods section of the paper, with an introductory paragraph at the beginning of the section introducing the EDT concept in the context of our study, which will help clarifying this point.

Line 320 – The thinning data still overlaps within uncertainty and remains stratigraphically consistent. The biggest issue is the disagreement with the 10Be ages.

We will modify this sentence to highlight the point made by reviewer 1: "*[10]Be ages show a general decrease with elevation, in agreement with progressive ice thinning along a deglaciation profile in the high Alps during the Late Glacial. This trend is less evident for the apparent [3]He ages, which overlap significantly within uncertainties above ~2200 m a.s.l. (Fig. 2).*" (line 362-364)

Line 323 – There is a similar pattern of 3He retention for both locations for elevations greater than 2150 m asl. Could it be discussed how this is, or is not, relevant?

We agree that, although significantly overlapping within error, $^{3}$He/$^{10}$Be age exposure ratios above 2150 m a.s.l. seem to decrease with elevation in both sites. However, because this trend remains within uncertainties, we decide to not further discuss this point.

Line 330 - *these Corrected

Line 337 – It might help to clarify that the approximate differences in EDT between high and low-elevation samples are determined from the peak value for each probability distribution.

Following the suggestion of Reviewer 1, we will modify the sentence as follows "*[…] they clearly depart from paleoIsoEDTs obtained from the low-elevation/Holocene sample(s), by ~6° (MBTP site) and ~18°C (GELM site) based on the peak values from the obtained bimodal probability distributions (Figs. 5b, e)*" (line 379-381).

**Referee #2 (Samuel Niedermann)**

General comment
The authors have measured cosmogenic $^3$He in quartz from two Alpine altitude profiles, with the aim to study paleo-temperature conditions in these areas that were gradually deglaciated after the LGM. They did the diffusion experiments that are required for a correct interpretation of the data and applied a lot of care in devising their experiments and model calculations, but obtained results that are difficult to interpret because temperatures recorded by cosmogenic $^3$He are generally lower than expected based on the models. They discuss possible explanations for the discrepancy, but cannot give a final answer. Nevertheless, I think this is an important paper as it shows the difficulties (but also the potentials) involved in the rather new cosmogenic $^3$He paleothermometry method and may thus prevent other scientists from misinterpreting their own data. With regard to the methods used, there is one thing that was not optimal in my opinion: Obviously, the helium analyses were confined to $^3$He (at least nothing else is reported). Measuring 4He also would have revealed any possible contributions of nucleogenic 3He, which may lead to an overestimation of cosmogenic 3He as argued in more detail below.

For the most part, the paper is written clearly and concisely. The English is generally fine, but suffers from quite a lot of small minor defects such as missing articles etc., which is a bit annoying given the fact that at least three co-authors are native English speakers. I have tried to point these out in the manuscript. Anyway, I recommend this manuscript for publication in Geochronology after minor revision has taken account of the specific and technical comments given hereafter.

We thank Dr. Niedermann for his positive appreciation of this study, his thorough reviewing of the manuscript and his very constructive comments.
We did measure $^4$He in addition to $^3$He; we address this point further below in the corresponding specific comment ("lines 172-182" and "lines 523-537").
All the edits directly indicated as comments in the pdf manuscript attached with the submitted review will be implemented. Likewise, all the suggested or requested text/figure modifications listed in the specific comments below were carefully considered, and we will implement most of them (cf. detail below).

Specific comments:
lines 172-182, the authors briefly describe the methods used for 3He determination. Surprisingly for me, they don't mention anything about 4He, and because there are no 4He or 3He/4He data in the tables either, I assume they have not even checked for 4He concentrations (otherwise those data should be given). This is a pity in the context of the discussion that many samples seem to contain more cosmogenic 3He than expected (see below).

We did measure the amount of $^4$He released by our natural samples during the two heating steps at 800 and 1100°C. The amount of $^4$He (in atoms; together with the amount of $^3$He for comparison) released during the 800°C heating step will now be reported in an additional table in the Data Supplementary (new Table S3).

While there was often some $^4$He released during the 1100°C step, no $^3$He above the blank level was ever measured during this heating step. Based on experiments quantifying helium diffusion kinetics presented both here and in previous works (e.g., Tremblay et al., 2014b), we argue that the $^4$He released during the 1100°C step is unlikely to come from the quartz itself but from mineral inclusions. Therefore, we will only report $^4$He and $^3$He amounts for the 800 ºC heating step. Furthermore, because it is evident that we did

not fully degas the $^4$He present in each aliquot despite fully degassing the $^3$He, we will report $^4$He measurements as amounts (atoms), rather than concentrations (atoms/g).

Please refer to our answer below (specific comment "line 525-537") for further discussion about a possible contribution of nucleogenic $^3$He, using the information from the measured amount of $^4$He released during the 800°C heating step.

Line 177: Why were blanks (only) measured at room temperature? Did you ever check whether they remain the same at higher temperatures up to the level used in the experiments? Otherwise that assumption seems a bit optimistic.

During a diffusion experiment, it is not possible to measure high-temperature blanks on the PtIr packet because this would also heat the sample and therefore ruin the step-heating nature of the experiment. However, at the end of the experiment when the sample has been fully degassed, we in essence do measure high-temperature blanks on the PtIr packet. We find that the $^3$He signals in these high-temperature 'blanks' are equal to those measured in the low-temperature blanks. Furthermore, in previous step-heating experiments we have heated empty PtIr packets from the same manufacturer and batch and found that they degas no helium above blank level. Aside from the PtIr packet and the sample inside of it, no other part of the diffusion cell apparatus is heated by the diode laser in our experimental setup.

In lines 523-537 the authors discuss a possible contribution of nucleogenic $^3$He in the samples that has not been accounted for. They conclude that it "does not contribute significantly", but their arguments are not very convincing because they have (obviously) not measured $^4$He. It is known that purely radiogenic/nucleogenic He is characterized by typical $^3$He/$^4$He ratios of ~$(2\pm2)\times10^{-8}$. Thus, if the measured $^3$He/$^4$He ratios in their samples were on the order of $10^{-6}$ for example, the nucleogenic $^3$He would be in the percent range at most and could safely be neglected. In my lab, I have measured several samples from similar settings (Mont Blanc area as well; work in progress) using two extraction steps of 600°C and 1000°C. While the 1000°C steps showed ratios in the $10^{-8}$ range as expected for radiogenic/nucleogenic He, the ratios in the 600°C steps varied from ~$6\times10^{-8}$ to several times $10^{-6}$, implying that the nucleogenic $^3$He contribution is not always negligible even in the low temperature step. Since the authors used 800°C as the first heating step, this is even more relevant for their samples. Their arguments for negligible nucleogenic $^3$He are not very powerful. Assuming the same diffusion characteristics for nucleogenic and cosmogenic $^3$He is probably not valid, since nucleogenic He is produced at different places in the minerals (namely, where Li is found, which is probably concentrated in mineral or fluid inclusions rather than in the quartz crystal itself). The fact that nucleogenic He is still degassed above 600°C while cosmogenic He is not shows on its own that the diffusion characteristics are different. Also, comparing the production rates of cosmogenic and nucleogenic He is not meaningful at all, since nucleogenic He has been produced over many millions of years, compared to the ~10 ka production of cosmogenic He. I don't say that the overabundance of cosmogenic $^3$He in the authors' samples is indeed due to nucleogenic $^3$He, but because of the lack of $^4$He data it is difficult to rule it out.

As mentioned in our response to the specific comment lines 172-182, we did measure the amount of $^4$He released by our natural samples and will report this information for the 800°C heating step in the newly added Table S3 in the revised Data Supplementary).

Dr. Niedermann is correct that nucleogenic $^3$He derived from thermal neutron capture on Li is, presumably, present in our samples in some amount greater than zero. For background, the source of

thermal neutrons for this reaction is decay of U and Th distributed throughout the whole rock, so the production rate of $^3$He from Li per unit Li is the same in quartz and in any hypothetical inclusions that are more retentive than quartz. Thus, the distribution of nucleogenic $^3$He from this process reflects the distribution of Li.

Dr. Niedermann hypothesizes that Li may be concentrated in retentive inclusions. In this case a significant amount of nucleogenic $^3$He would be concentrated in these inclusions, and some of it would be released in the first heating step. If this is true, just as we see a large fraction of total $^4$He in the higher-temperature heating steps, we would expect to detect $^3$He above background in the higher-temperature heating steps. We did not see this in any sample. Instead, $^3$He signals in all 1100ºC steps measured were indistinguishable from blank. Therefore, retentive inclusions may contribute a nonzero amount of nucleogenic $^3$He, but this amount is not detectable.

The alternative hypothesis that would allow for a significant contribution of nucleogenic $^3$He in the low-temperature heating step would be that Li is present in the quartz or in inclusions with retentivity that is similar to or lower than quartz. If this were the case, the maximum concentration of nucleogenic $^3$He that could be observed in our low-temperature heating steps would be limited by its production rate and the diffusion kinetics of quartz. The nucleogenic $^3$He production rate in quartz for a granite with average composition (3.5 ppm, U, 16 ppm Th) and quartz with 70 ppm Li (maximum concentration measured for the Aare granite; Schaltegger and Krähenbül, 1990) is approximately 1 atom/g/yr. For the diffusion kinetics measured in our quartz samples, and assuming a 0.850-mm radius quartz grain (in between the two median grain sizes for our samples) at 0 ºC, the steady state nucleogenic $^3$He concentration would be at most $2.8 \times 10^4$ atoms/g, which is two orders of magnitude less than our measured concentrations of cosmogenic $^3$He and therefore undetectable.

The only way this reasoning could truly be tested would be to obtain samples of the same lithology from the deep subsurface that have not been exposed to the cosmic-ray flux. Nonetheless, the above reasoning strongly indicates that significant nucleogenic $^3$He is not present in our quartz samples. Thus, we conclude that the contribution of nucleogenic $^3$He to our measured $^3$He concentrations in the first heating steps is at or below the level of measurement error and therefore undetectable.

Even if we ignore all of the above reasoning and estimate a non-cosmogenic $^3$He amount using a $^3$He/$^4$He ratio for a radiogenic/nucleogenic component of $\sim(2\pm2)\times10^{-8}$, as quoted by Dr. Niedermann, we find that this amount of non-cosmogenic $^3$He is not significant. Specifically, the calculated amount of non-cosmogenic $^3$He released during the 800ºC step is in all cases smaller than the uncertainty of the total $^3$He amount measured. Finally, we observe no correlation between amounts of $^3$He and $^4$He released in the 800ºC heating steps in samples from either transects (see fig. RN1 below), and we would expect to observe a correlation if there were a nontrivial amount of nucleogenic $^3$He present in our samples.

[Figure]

Figure RN1: Amounts of $^3$He and $^4$He released in the 800ºC heating steps in samples from the MBTP (left) and the GELM (right) transects

We will now modify the main text in order to consolidate our argument for negligible nucleogenic $^3$He as follows:

*"[…] However, we think it is unlikely that there is significant nucleogenic $^3$He in our samples for several reasons. First, the $^3$He/$^4$He ratios we measured during the 800°C heating step are on the order of $10^{-6}$ to $10^{-7}$ (Table S3). This is more than an order of magnitude above the $^3$He/$^4$He ratio of ~$10^{-8}$ expected from U/Th decay and $^6$Li neutron capture (Niedermann, 2002). Second, no $^3$He above the detection limit was measured in the 1100 ºC heating step despite nontrivial amounts of $^4$He being released in this step. This indicates that retentive mineral and fluid inclusions, if present in the samples, are not contributing a significant amount of non-cosmogenic $^3$He to the measured $^3$He amounts. Third, based on the diffusion kinetics of $^3$He in quartz, we anticipate that any nucleogenic 3He produced in the quartz itself over geologic timescales will be diffusively lost before the sampled rock surface is exhumed at near-surface temperatures. Furthermore, the production rate of nucleogenic $^3$He is low compared to the cosmogenic production rate of $^3$He. We do not have direct data of major and trace element for the MBTP and GELM samples in order to calculate the nucleogenic $^3$He production rate directly. However, we can say that a rough maximum estimate for the production rate of nucleogenic $^3$He in the GELM samples is ~1 at/g/yr, which is based on a maximum Li concentration of 70 ppm for the Aare granite (Schaltegger and Krähenbühl, 1990) and the production rate estimate of Farley et al. (2006) for an 'average' granite. This is 0.3% of the local, scaled production rate of cosmogenic 3He for sample GELM18-9, which has the lowest cosmogenic $^3$He production rate of all of our samples. Given this maximum production rate estimate for nucleogenic $^3$He, and using the Holocene-calibrated diffusion kinetics for our samples, we estimate that the maximum steady-state concentration of nucleogenic $^3$He is 2.8 × 10$^4$ atoms/g, which is two orders of magnitude smaller than the measured 3He concentrations in our samples and well within the uncertainties of those measurements. It is therefore unlikely that not correcting for nucleogenic $^3$He affected our modeled ΔEDTs in any significant way."* (line 579-595)

*Technical comments:* (numbers refer to line numbers in the manuscript)
5-10 Please give zip codes of European cities before the city name (without comma), as customary here.

We will modify the text as suggested.

19 Should be "... in quartz from the Mont Blanc site and complex Arrhenius behavior in quartz from the Aar site...", as "behavior observed from the Aar site" is an odd wording.

We will modify the text as suggested.

31 Tremblay et al. 2014a or 2014b?

We will specify "Tremblay et al., 2014a".

57 It is odd to say "glaciers retreated quickly behind the Little Ice Age moraines" when considering a time long before these moraines were there! Change to something like "behind the position were today the Little Ice Age moraines are located".

The sentence will be modified as follows: "*[…] glaciers retreated quickly behind the position where the Little Ice Age moraines are located today, […]*". (line 57-59)

64 I assume this should rather read "High-resolution δ18O in Alpine speleothems …"

We will modify the text as suggested.

70 "which resulted in … North Atlantic atmosphere patterns"??? Do you mean in variable atmosphere patterns? Then you should repeat that word (it's in singular in the first part, so it doesn't seem to belong to the plural term in the second one).

This sentence will be reworked for clarification as follows: "*Indeed, major climate forcing components, such as ice sheet extent, atmospheric greenhouse gas concentrations, and changes in ocean circulation, also underwent large-scale changes between the LGM and the Holocene transition (Clark et al., 2012). This resulted in variable atmospheric circulation patterns (Eynaud et al., 2009) and variable latitudinal temperature gradients (Heiri et al., 2014b) in the North Hemisphere during this period.*" (line 68-72)

77 There is no Bartlein et al. 2014 in the reference list (only B. et al. 2010).

We thank Dr. Niedermann for pointing out this mistake. We will correct this reference in the main text and in the reference list, with Bartlein et al. (2011).

79 Please explain ELA for those readers who are not familiar with that term.

The explanation of ELA will be implemented in the text as follows: "*[…] reconstructed ice limits and estimates of paleo-Equilibrium Line Altitude (ELA; i.e., the elevation at which annual net ice budget in a glacier equals zero […]*". (line 79-80)

132 "from the exact same locations previously collected" seems to imply you sampled surfaces from which some rock had already been knocked off before, i.e your surfaces would have been covered until a few years ago. I'm sure that's not what you did, but please clarify it!

Samples for [3]He analyses were collected at the same sample locations previously reported in Lehmann et al. (2020) and Wirsig et al. (2016b). We will clarify the location of [3]He samples relative to the previous [10]Be sampling points as follow: "*In this study, new samples were collected for [3]He experiments from the same rock surfaces and locations as the sampling sites previously collected for [10]Be dating by Lehmann et al. (2020; MBTP profile, samples MBTP18 -1, -2, -11 and -9, n=4) and Wirsig et al. (2016b; GELM profile, samples GELM18 -12, -1, -5, -11, -6 and - 9, n=6; Fig. 1, Table 1)*". (line 140-143)

Fig. 2: On the y axis labels, remove dot after m because the symbol m for meter is never written with a dot. In the caption, it is confusing to write (a-c), (b-d) which seems to mean a to c and b to d, while obviously a and c, b and d is meant. So change to (a, c), (b, d).

Changes requested for the y axis labels on the figure and in the caption will be implemented.

Table 1: Lehmann et al. 2019 and Borchers et al. 2016 are not found in the reference list (but L. et al. 2020, B. et al. 2015). Give details about the method to estimate Mean Annual Rock Surface Temperatures. Please explain EDT or, at least, refer to the text section where it is explained.

We thank Dr. Niedermann for pointing out the mistake regarding the references. We will correct the main text and reference list in order that references to Lehmann et al. (2020) and Borchers et al. (2016) will be consistent.

Effective Diffusion Temperature will now be written in full in the caption.

The method to estimate (modern) Mean Annual Rock Surface Temperatures and associated Effective Diffusion Temperature is described in section 3.1 ("Effective Diffusion Temperature estimates"; which will be moved to section 3.5 in the revised manuscript, cf. answer to general comment from reviewer 3 for Analytics approach). To avoid confusion and following Dr. Niedermann's suggestion, we will explicitly refer to section 3.5 in the caption of Table 1 for the detailed explanation regarding modern MARST and EDT estimates.

184 Tremblay et al. (2021): Only Tremblay (2021) in reference list.

We will correct the reference to Tremblay et al. (2021) throughout the text and in the reference list.

191 Is this the spallogenic or total 10Be production rate?

This is the spallogenic 10Be production rate. We will state this in the main text.

192 Borchers et al. 2016 see above (Table 2)

We will correctly report Borchers et al. (2016) in the reference list.

243 What is "0.5 increment"? Do you mean increments of 0.5 kJ/mol? If so, you must give the unit!

We will add the unit kJ/mol.

244 Baxter et al. 2010 is not in reference list (only Baxter 2010).

We thank Dr. Niedermann to point out this mistake. The correct reference is Baxter (2010) and will be implemented throughout the main text.

Fig. 3: I can't see any gray lines, just gray areas between red and blue lines! This whole figure is very confusing and difficult to decipher, and the explanations in the caption do not help a lot. I ask the authors to think about how this figure could be improved, e.g. by showing several panels per location, and how it could be explained better.

We will modify the figure 3 by:
- thinning the lines to make them more visible in the main Arrhenius plots (a, b)
- adding 3 panels (c, d, e) showing forward $^3$He simulation runs using the different diffusion kinetic parameter solutions fitting the laboratory heating step degassing experiments (grey lines), and testing which of those parameters allow to reproduce $^3$He concentrations observed from samples exposed during the Holocene period only (blue lines). For the MBTP site (b), only one sample (MBTP18-9) was used for the Holocene calibration step, while for the GELM site, two Holocene samples were available (GLM18-6 and -9).

The figure caption will be modified accordingly, as well as references to the different panels throughout the text.

284 Related to the above, I don't understand how the parameters Ea and ln(D0/a2) can be obtained from Fig. 3.

Please refer to our detailed answer to the comment above regarding the modifications we will conduct on Figure 3.
We believe that by adding these three panels (c, d, e) in Figure 3 will permit to better illustrate the Holocene calibration step applied in addition to the MDD analyses of laboratory data from heating-step degassing experiments. In addition, we will explain in the caption of the figure how the parameters $E_a$ and ln($D0/a2$) are derived from the Arrhenius plots: "*(Ea is proportional to the slope of each line, and ln(D0/a2) is given by the intercept)*" (line 309-310). Lastly, we will rework the text in the subsection "Diffusion kinetics determination" to further clarify this multi-steps procedure that allowed us to determine the final (Holocene calibrated) diffusion kinetics parameters used in the numerical simulations thereafter.

Table 2: 4th column from left, MBTP d3: Range 8.67 to 10, is that 8.67 to 10.00? If so, don't omit the .00, because otherwise it looks like this value was much less precisely determined than the other ones.

10.00 will be added in Table 2.

Fig. 4: At such a limited range of values (~2000-20,000), I don't see why a logarithmic scale is used for the y axis.

We will modify figure 4 by applying a linear scale to the y axis.

341-342 I assume what you mean is "… from the highest sample (…) partially overlaps that from MBTP18-9 within uncertainty."

We will modify the sentences as suggested: "*…from the highest sample (…) partially overlaps that from MBTP18-9 within uncertainty*". (line 385)

Fig. 5: What is the difference between the solid and dashed green lines in panels e and f?

There are two Holocene samples along the GELM deglaciation profile. The two green lines represent the two modern EDTs associated with those samples, however because only one modern EDT could be used as reference for applying the lapse rate correction to temperature presented in panel f, we made a distinction: the modern EDT used as reference for applying the temperature correction is represented by

the solid green line (and associated with the sample with a black asterisk on panel f). The other modern EDT represented by a dashed line was not used for the temperature correction.

This will now be specified in the caption as follows: *"Green solid and dashed lines are modern EDTs for Holocene samples, with the solid line indicating the modern EDT taken as reference for the lapse rate correction".* (line 391-392)

374 Instead of "experiments", which may imply something physical, better write "model runs" or similar.

We will replace "experiments" by "model runs". In addition, in other places of the main text were the term "experiments" was used alone for designating numerical model simulations, we will change the wording and use either "numerical simulation" or "model run".

487 Prud'Homme or Prud'homme as spelled in reference list?

We will spell it *"Prud'homme et al., 2020"* in the main text, as spelled in the reference list.

514 Feldspar is not a mineral that is retentive for $^3$He; actually it is even less retentive than quartz! I don't think Masarik and Reedy have based their numbers on feldspar.

The other retentive mineral referred to was indeed not feldspar but olivine. This will be corrected. In addition, we will modify the way the references are ordered in this sentence to improve clarity: *"Alternative approaches using artificial targets (e.g., Vermeesch et al., 2009) or scaling $P_{3He}$ measured in retentive minerals (i.e., olivine; e.g., Cerling and Craig, 1994; Goehring et al., 2010) have hence been used. […] a ~10% higher $^3$He production rate has also been proposed from olivine $^3$He measurements scaled to quartz (e.g., Masarik and Reedy, 1995; Ackert et al., 2011)".* (line 565-568)
The reference list will be updated accordingly.

533 Farley et al. 2006 is not in reference list.

The reference will be added in the reference list.

547 Liu et al. 2021 is not in reference list.

The reference will be added in the reference list.

557 Berger and York 1981 is not in reference list.

The reference will be added in the reference list.

596, 601 Baxter et al. 2010 is not in reference list (only Baxter 2010).

The reference to Baxter (2010) will be corrected throughout the main text.

Data supplementary:
Fig. S1 Please use consistent sample labels throughout the manuscript. Here, GELM18-1 and GELM18-6 is written in the legend but GELM18-01, -06 in the figure caption.

The sample labels in the figure are the right ones. Sample names in the figure caption will be corrected accordingly.

Tab. S1, S2: The uncertainties given for 3He are unrealistically small, in some cases <1 permil! This is obviously nonsense, no mass spectrometer is able to measure as precisely as that. I assume the authors only considered the statistical error of the measurement, disregarding other error sources such as mass spectrometer sensitivity or linearity. Even if the systematic errors (such as the uncertainty of the He concentration in the standard that is used for sensitivity determination) are not taken into account, which makes sense in such a step-heating experiment (but should be noted), short-term variations of the sensitivity or deviatons from the calibrated or assumed linearity of the mass spectrometer signal will be in the percent rather than the permil range.

Uncertainties associated with the $^3$He amount (in atoms) released during our diffusion experiments and reported in the Tables S1 and S2 were indeed calculated without taking in account short-term mass spectrometer sensitivity variation/deviation from assumed linearity.
Following our initial approach, reported $^3$He uncertainties for step-heating experiments varied between:
- for GELM18-1 (Table S2; which will become Table S5 in the revised version of the Data Supplementary): 0.3 to 5%. We do not assess these uncertainties to be unreasonably low considering the large amount of $^3$He released (in the range of $10^7$ to $10^8$ atoms per heating step). It is worth noting that these $^3$He amounts released from the proton-irradiated quartz grains are much larger than the natural amounts measured in unirradiated aliquots, so the uncertainties associated with the former measurements tend to be smaller.
- for MBTP18-9, there was an error in the reported uncertainties associated with the conversion of mols into atoms, leading to unreasonably low uncertainties (<0.1% for steps with $^3$He amounts of $10^7$ to $10^8$ atoms). After correction of this mistake, uncertainties range between 0.4 to 5% for $^3$He release amounts of ~$10^7$-$10^8$ atoms per step, and are hence assessed as reasonable. The supplementary table will be updated accordingly (Table S4, in the revised version of the Data Supplementary). The reported mistake didn't affect our diffusion kinetic analysis with the MDD model. We thank Dr. Niedermann for helping us to identify this error in our reported data.

Following the suggestions of Dr. Niedermann, we calculated the uncertainties including short term mass spectrometer sensitivity deviations from linearity. The calculated uncertainties range between:
- For GELM18-1:  5 and 8%
- For MBTP18-1: 4 and 7%
We will also report these uncertainties in Tables S4 and S5 in the updated Data Supplementary.

Tab. S3: Similar to the 3He uncertainties in Tab. S1 and S2, here the 10Be uncertainties in the permil range are unrealistic. Again, it looks like only the 10Be/9Be measurement error was taken into account, without considering the uncertainty of the standard or carrier for example.
Footnote 2 should refer to Table S4 rather than Table S2.

We acknowledge a mistake in the uncertainties for the $^{10}$Be concentrations initially measured by Lehmann et al. (2020) and Wirsig et al. (2016b) and which were incorrectly reported in this table.
We will correct this in the updated Table (Table S1) of the Data Supplementary, such that the reported $^{10}$Be concentration uncertainties will be in the percent range (between 3 and 8%).
Similarly, final uncertainties for the re-calculated $^{10}$Be ages will also be corrected (age uncertainty increase by 0.1 to 0.3 ka) in the updated Table 1. This error however does not affect our numerical simulations of $^3$He diffusion for varying time-EDT scenarios.

At last, [10]Be error bars in figures 2, 5, 7 and 10 will also be corrected accordingly.

The reference in the footnote will also be corrected and refer to the correct table (Table S2 in the updated version).

Tab. S3, S4: In sharp discrepancy from the former two tables, 3He errors are very high (on the order of 10%) here. While this may be realistic, the reason for the much higher uncertainties should be explained even when those in Tables S1 and S2 have been adjusted to a more realistic level.

We explain the difference in relative [3]He uncertainties between the diffusion kinetics measurements (Tables S1, S2; Tables S4, S5 in the revised version) and the natural [3]He measurements (Table S4, S5; Tables S1, S2 in the revised version) by the significantly larger amount of [3]He contained in the proton-irradiated samples used for diffusion kinetic experiments, compared to the natural samples.
When both data sets are evaluated as total amounts of [3]He in atoms, the measured [3]He amounts are $10^7$–$10^8$ atoms per heating step for the proton-irradiated samples and around ~$10^5$ atoms for the natural samples (except for GELM9, with ~$10^4$ atoms). On the other hand, the absolute uncertainty based on counting statistics in both cases is $10^4$–$10^5$ atoms. Therefore, the relative uncertainty appears significantly higher for the natural [3]He measurements than for the diffusion kinetics measurements carried out on proton- irradiated samples.

The 3He concentrations and their uncertainties are given with unreasonable precision, i.e 6 significant digits for the errors. It is inappropriate (and confusing) to give more than two significant digits for an uncertainty, because uncertainties are not precise numbers but just represent probabilities. Therefore, values such as 520,929 should be rounded to 520,000, and of course the corresponding values should always be given with the same precision as the uncertainties. The first entry in Table S3 would thus be 6,400,000 ± 520,000, or better readable (and easier to round in Excel, e.g.) 6.40 ± 0.52 in units of 106 at/g.

Tables S3 and S4 (Tables S1, S2 in the revised version) will be corrected with correct significant figures and with [3]He concentrations reported in $10^6$ at.g$^{-1}$, as suggested.

I wonder how the average values from the three replicate measurements were calculated. For three measurements agreeing within error limits, as seems to be the case looking at Table S4, an error-weighted mean could be calculated. Obviously, the authors calculated an unweighted average instead and, very strangely, used the average of the errors as the error of the average. I don't think this is a statistically correct way.

The unweighted mean was indeed taken to estimate the final [3]He concentrations. The unweighted means and error weighted means (EWM) do not differ significantly (by 0.1 to 3.5%), due to the fairly constant relative error between the replicate measurements.

Regarding the final uncertainty, we indeed took the average of the 1σ errors recorded from the three replicate measurements per sample.
We recognize that a more conventional practice would have been to take the standard deviation, allowing us to estimate averaged [3]He concentrations to a 68% confidence level. This would have led to differences in the relative error of the mean [3]He concentrations of up to 10% for all the samples but one (GELM18-9, Holocene sample), with no systematic decrease or increase in the percent error.
Using the standard deviation instead of the average error would affect:

- the range of solutions of diffusion kinetic parameters that match the [3]He concentration from the Holocene samples during the Holocene calibration step (cf. section "Diffusion kinetics determination"). However, this will not affect the final set of diffusion kinetic parameters used for this study, which was selected based on the closest match with the average [3]He concentration.
- the final results from our forward [3]He simulations for varying time-EDT scenarios (two last sets of experiments described in section 4.3, cf. figs. 7 and 8), as simulated [3]He concentrations were compared with observed [3]He concentrations with uncertainties. However, this does not change significantly our results, as shown in the figure below (fig. RN2), where we compare the solutions of time-EDT scenario explaining our observed [3]He concentrations for the MBTP samples obtained using [3]He errors either derived from the average error (left panel, as done in our study) or from the standard deviation. Scenarios with significantly colder past temperatures than expected and maintained until very recent times are still necessary to explain our observed [3]He concentrations.

[Figure]

Figure RN2: Solutions of time-EDT scenario explaining our observed [3]He concentrations for the MBTP samples obtained using [3]He errors either derived from the average error (left panel, as done in our study) or from the standard deviation (right panel)

To limit any confusion, we will 1) clearly specify in the footnote of Table S1 in the updated Data Supplementary how [3]He concentrations and uncertainties have been calculated; 2) modified the heading of the column where final [3]He concentration errors are reported with the term "Avg. Unc." for average uncertainty (instead of "±1σ" previously used); 3) added a column with the standard deviation.

**Anonymous Referee #3**

**Summary:** "Diffusion properties of cosmogenic $^3$He in quartz can be quantified to reconstruct the evolution of past in-situ temperatures from formerly glaciated areas." For this study, authors apply cosmogenic 3He palaeothermometry on rock surfaces exposed from the LGM–Holocene from two deglaciation profiles in the European Alps. **Results:** "Lab experiments indicate variability in $^3$He diffusion kinetics between the two sites." The authors interpret this as the presence of multiple diffusion domains. "Predictive simulations indicate that $^3$He abundance in all the investigated samples should be at equilibrium with present-day temperature conditions - but measured natural $^3$He concentrations in samples exposed since before the Holocene indicate a thermal signal colder than today. This cannot be explained by realistic post-LGM mean annual temperature evolution in the Alps. **Hypotheses/interpretations:** One hypothesis is that the diffusion kinetics and MDD model applied may not provide sufficiently accurate paleotemperature estimates in these samples. Alternatively, the $^3$He abundance may reflect complex geomorphic/paleoclimatic evolution with much more recent ground temperature changes associated with the degradation of alpine permafrost."

**Overall comments:** This is a thorough manuscript and important study that provides detailed information on applying relatively novel methods to deconvolve temperature histories from formerly glaciated areas. The authors do a nice job talking through their interpretations, and all of their interpretations are valid. The amount of data collected and reported is impressive. **My main concern** with this paper is the sheer amount of information that is reported and the delivery of details that make it hard to follow at times. As a result, key takeaways are not obvious, and this diminishes the paper's impact. For example, I noticed throughout the manuscript the author put statements in parentheses within sentences, and I felt this was quite distracting. More work into simplifying those statements and perhaps removing the text within the parentheses will help streamline the paper.

We thank the reviewer for their interest in the study and overall positive response.
We will improve the readability and clarity of the manuscript by (1) modifying the structure of some paragraphs, (2) reorganizing and changing the titles of some sections, and (3) rewriting sentences to avoid parenthetical statements where possible.
Please see our responses to the line and section comments below.

*Most of my comments below are my recommendations for how the authors can better streamline the paper's structure. If a bit of work is put in on improving the structure of this manuscript, it will be much stronger.

**Comments by section:**
**1 INTRODUCTION**
General comment: The introduction does a nice job setting up background information for this study. Thank you.

**Line 48:** suggestion to change "thanks to" to "due to."

We will modify as suggested.

**Line 59**: I'm a bit confused by the beginning of this sentence, "*the recorded Alpine sequence*" What exactly do you mean by it? Which "recorded Alpine glacial sequence?" Perhaps reword to say: "Alpine glacial history is consistent with polar ice oxygen isotope records."

We will modify this sentence as follows: "*The timing and pattern of paleoglacier variations in the European Alps are consistent with polar ice oxygen isotope…*". (line 60)

**Line 67:** A suggestion to change "inappropriate" to "not viable."

We will replace "*inappropriate*" by "*not valid*". (line 68)

**Line 91:** Suggestion to change "today" to "presently," and remove "hence." So, it would read: "*Presently, there is a crucial lack of...*"

We will remove "hence".

**Line 93:** Suggestion to reword to: **"***In this study, we attempt to reconstruct paleotemperatures in the high alps during the Late Glacial Period, specifically between the LGM and YD, by***…"**

We will modify the sentence as follows: "*In this study, we attempt to reconstruct paleotemperatures in the high Alps during the Late Glacial by …*". (line 96)
Information about which period the Late Glacial corresponds will be provided earlier in the introduction: "*Similarly, little is known regarding climatic conditions during the Late Glacial period between the LGM and the YD, …*".  (line 85-86)

**Line 94-95:** Perhaps add a brief statement at the end of this sentence for what the benefit of "exploiting the open-system behavior of cosmogenic 3He in quartz minerals" is? In other words, why is it important to "exploit the open system behavior?" As of now, this statement seems like jargon, and elaborating on your meaning behind this will fix the ambiguity.

We will replace "open-system" by "diffusive". The benefit of this property (diffusion of $^3$He in quartz at Earth surface temperatures) is described in the sentence immediately following this one, which we will modify to be more explicit: "*Using forward models of cosmogenic $^3$He production and thermally-activated diffusive loss, quantitative constraints on the thermal history of an exposed rock surface can thus be inferred from the difference between surface-exposure ages derived from the diffusive $^3$He system and from a cosmogenic nuclide that does not experience diffusive loss (Tremblay et al., 2014a, b; 2018)*". (line 98-101).

**Line 95:** I think this manuscript would benefit if you briefly describe what you mean by "predictive models."

We will replace the word "predictive" here and elsewhere with the word "forward," as this is a more intuitive and accurate description of the modeling that was done.

**Line 98:** "Leaky" is a bit confusing. For readers that are not experts in 3He palaeothermometry, what do you mean by "leaky"? Be more specific please. (I think you're referring to open-system behavior? Perhaps just say that instead of leaky if so).

We will replace *"leaky"* by *"diffusive"*.

**Line 102:** I think you need to define "*a priori*," I'm not sure what you mean by this, it feels out of place.

We will delete this word.

**2 STUDY SITES AND SAMPLE MEASUREMENTS**
**General comment:** Permafrost degradation is one of your hypotheses to explain your results, yet there is no mention of permafrost extent in the "settings/study sites" section of your paper. I think you should mention modern and/or past permafrost extent in your study area.

We will add the following brief description of the present-day permafrost distribution in both study areas: *"Continuous permafrost is expected above ~3000 m a.s.l in the north faces of the Mont Blanc massif (permafrost index ≥0.9; Magnin et al., 2015a) but can be found more discontinuously down to 2300 m a.s.l. (permafrost index ≥0.5) and as low as 1900 m a.s.l. in especially favorable conditions (permafrost index ≥0.1). Along the Gelmersee ridge on the western side of the Haslital Valley, continuous permafrost is expected above ~2700 m a.s.l., with sporadic patches down to ~2150 m a.s.l. (Boeckli et al., 2012b)."* (line 126-130)

**Line 125:** Figure 1 is really great! I appreciate the visual. Thanks!

**Line 138:** A suggestion for Figure 2 is to add a key box to panels b and d that state "3He/10Be" with a yellow circle symbol. I realize it's in the x-axis label, but consistency with the keys will be more apparent to readers. Also, a suggestion to not use a dash ("-") but instead a comma in your caption because I think of "a-c" as being "a, b, and c... but you only mean a & c.

While more visually consistent with panels a and c, we decided to not add additional key boxes in panels b and d for Figure 2, as this would be redundant with the x-axis label (on the contrary to panel a and c where the key boxes really bring an additional information). We will however modify the x-axis label to be more explicit with "$^3$He exposure age/$^{10}$Be exposure age". We will also replace the dashes ("-") by commas in the figure caption.

**Lines 148-150:** I like this set up: *"3He palaeothermometry requires at least two additional pieces of information. **First**, predictive models of thermally-activated 3He diffusion rely on quartz sample-specific 3He diffusion kinetics parameters…**Second**, measurement of the total natural cosmogenic 3He accumulated in the quartz sample permits us to estimate the loss by diffusion..."* I think you should incorporate more summation sentences like this within your manuscript to help streamline the flow.

We agree with the reviewer, and will work on more statements like this (see next responses).

**3 ANALYTICS APPROACH**
**General comment:** As someone who is not an expert on 3He palaeothermometry, I'm a bit confused of the significance of this section. Why do you need an analytics approach? Is it to perform quality control metrics on your data? Or do you need to run these Matlab codes to actually produce your results? Please start this section with 1-2 sentences briefly explaining why you need to run these Matlab codes. *In the previous section, you just said how you need "*predictive models of thermally-activated 3He diffusion**...**" is this why you're running the Matlab code? If so, a suggestion is to start this section with, "*To determine predictive models of thermally-activated 3He diffusion, we used Matlab codes…*"

We appreciate that it was difficult to follow the description of methods in our original manuscript. In response to this, we will significantly restructure this part of the paper to have a Methods section with subsections that have clear, descriptive titles. We will also add a synoptic summary at the beginning of the methods section which clearly delineates the different measurement methods and modeling methods used, what the purpose of each method is, and how each of these parts relate to one another. We will also remove the specific references to Matlab code, as this detail isn't particularly relevant.

**Line 186:** This is the first mention of "multi-diffusion domain (MDD)," but there is no explanation for what this is? Please provide a bit of an explanation for what MDD is.

Please refer to our answer above (comment: 3 ANALYTICS APPROACH General comment). We will explain what MDD is in both the synoptic summary at the beginning of the Methods section as well as in this subsection on the diffusion kinetics modeling (section 3.3 in the revised manuscript).

**General comment:** Once again, I think this section would benefit from an opening sentence briefly stating/reminding the reader of why you need to run these Matlab codes. For example, why do you need to determine 3He diffusion kinetics, and why do you need to numerically simulate 3He loss? *I see at **Line 300** you state "*to explore the theoretical sensitivity and potential variability of the MBTP and GELM quartz, we numerically evaluated the time required…*" Is this why you ran the Matlab codes? If so, please clarify this in section 3 above, instead of waiting to the results section to explain this.

Please refer to our answer above (comment: 3 ANALYTICS APPROACH General comment).

**Line 199:** Please be more specific with what you mean concerning "longer timescales." >1 year? Decadal? *I see at **Line 229** that you define "long-term" as "several years." Please add this clarification to Line 199.

We will specify that longer timescales mean 1 to $10^5$ years.

**Line 249:** You can remove the word "next."

We will make the change as suggested.

**4 RESULTS**
**General comment:** I think this section could be improved if you added an introductory paragraph summarizing all the different results you are going to report. For example, as it is now, you jump right into diffusion kinetics parameters, which is a bit confusing/makes me think these are the most important results, even though you delve into many more below this section.

We will add this introductory paragraph: "*First, we examine the characteristics of the $^3$He diffusion kinetics parameters we modeled for our quartz samples and explore the sensitivity of the $^3$He signal in those samples to Earth surface EDTs. We then present forward model results for the evolution of the cosmogenic $^3$He concentrations recorded along each deglaciation profile for two different sets of thermal histories. The first set of thermal histories we investigate assumes a constant EDT since the exposure of the sampled rock surfaces following ice retreat. We then investigate a set of more climatologically-interesting thermal histories, wherein a change in EDT occurs at some point during the exposure time of each sample.*" (line 319-324)

**Line 285:** The wording of "*and which in addition predict*" is confusing. Do you mean the range of diffusion kinetics parameters predict the observed natural 3He concentrations from the Holocene calibration samples? If so, please remove "and which in addition" and replace with "*which predict*" wording only.
We will modify the sentence as follows: *"Figure 3 shows the range of diffusion kinetics parameters (Ea and ln($D_0/a^2$)) that fit the laboratory stepwise-heating experiments (one representative sample for each site; Fig. 3a and 3b), and which permit us to reproduce the observed natural $^3$He concentrations from the Holocene calibration samples for a constant EDT equivalent to the modern EDT (Figs. 3c-e).*" (line 326-328)

**Line 315:** Figure 4 - a suggestion to make the diamond symbols transparent because the circles are often hidden behind them, and this would show their clear overlap. Also, are there errors of these isoEDT estimates?

We will make the scale of the y-axis in Figure 4 linear. Consequently, most of the symbols will not be overlapping. We will also put the circles on top of the diamond symbols, such that none are hidden where there is overlap.
There are no errors for this figure, as this figure presents illustrative synthetic tests where we varied the isoEDT and predicted the time it would take for the $^3$He concentration in quartz to reach steady-state, and for diffusion kinetics parameters were held constant.

**5 DISCUSSION**

**Lines 417-421:** These sentences belong in the results section; they distract from your interpretations. A suggestion is to move these statements to the above results section and **Line 434:** "*We attribute…*"

These two sentences summarize the results presented in details in the preceding result sections.
We find them helpful to start the discussion of the presented results in regard to the existing knowledge of climate evolution since the LGM in the European Alps. We therefore will keep them.

**Line 423:** Remove the word "furthermore," and add "set of" after "mean." So, it reads: "*Such large EDT differences would not be supported by any set of mean temperature reconstructions for the…*"

We will modify the sentence as suggested.

**Line 434:** Suggestion to reword this sentence to: "*We attribute our results of lower modeled 3He concentrations to observed 3He observations to the dampening effect…*"

We will rework this sentence as follows: "*We attribute the result of modeled $^3$He concentrations that are significantly lower than the observed ones to the damping effect …*" (line 479)

**Line 435:** What do you mean by the "dampening effect?" Perhaps clarify more, I'm not sure how "relatively stable mean temperature conditions similar to present day" relate to this.

We mean that the long exposure of quartz systems to temperatures similar to today during the Holocene period (~10 ka), will result in a partial to total re-adjustment of the $^3$He thermal signal, with little to no inherited signal (i.e., signal memory) from prior exposure to colder Late Glacial conditions. We will add a sentence with this explanation to the text (line 481-483).

**Line 436:** You can remove "first appears," and just say "*This hypothesis contradicts...*"

We opt to keep the word "appears." As explained in this paragraph, the proposed hypothesis is in fact consistent with our tests when the limited pre-Holocene samples exposure time is considered.

**Lines 459-464:** The parentheses are distracting in the reporting of your interpretation. I think you can reword this paragraph as: "*Even greater EDTs are needed to explain the observed GELM 3He concentrations using either diffusion kinetics approach, including temperature changes from 15-35ºC when using Holocene-calibrated diffusion kinetics and >40ºC when using laboratory diffusion kinetics.*"

We agree that the amount of details in this paragraph is overwhelming and might be confusing for the reader. We will simplify this paragraph and refer the reader to the supplementary material for further detailed information as follows: "*If instead we use diffusion kinetics solely derived from laboratory experiments without Holocene calibration (Fig. 3, Table 2), a ΔEDT of 15°C or greater is required to explain observed MBTP $^3$He concentrations for changes within the last $10^3$ years (Fig. S2b). Such large ΔEDTs are significantly greater than expected EDT variations from changes in mean annual temperatures and/or in annual/diurnal temperature amplitudes during the Holocene. Even greater ΔEDTs are needed to explain the observed GELM $^3$He concentrations using either diffusion kinetics approach (Fig. S3b).*" (line 503-508)

Then, remove the statements in the parenthesis and make **Line 464** its own sentence as: "*Both cases are clearly incompatible with…*"

We will modify as suggested.

**Line 466**: Suggestion to remove the word "Finally" at the beginning of this sentence.

We will modify as suggested.

**Line 470:** Remove "*(i.e., recent)*" statement, it is not needed.

We will modify as suggested.

**Line 479:** Suggestion to remove the words "*permit to,*" so it reads: "*For GELM samples, correcting modern/recent EDT does not reproduce the observed 3He concentrations…*"

We will replace "*does not permit to*" by "*is not sufficient to*". (line 530)

**Line 490:** Remove the parentheses in front of "contrary," and reword as two full sentences: "*…during glacier coverage. This is in contrary to 10Be, which would experience…*"

We will reword the section as two sentences:
"*[…] during glacier coverage, even at subzero temperatures and EDTs (Fig. S4). On the contrary, $^{10}$Be would experience only minor radioactive decay over 10-100 kyr timescales.*" (line 541-543)

**491:** Suggestion to remove the parenthesis and the word "hence" and just reword as: "*This scenario of inheritance and/or complex exposure history would result in…*". This streamlines your interpretation.

We will modify the text as follows:

"This scenario of inheritance and/or complex exposure history would result in […]. This scenario is also valid for post-LGM episodic coverage." (line 543-545)

**Section 5.3 (Line 538):** I'm a bit confused how this discussion differs from Section 5.1. Aren't the 3He diffusion kinetics and 3He thermal signal needed to interpret paleoenvironmental signals? Why is this discussion lumped in a different section?

We agree that the current structure of the discussion part and focus of the different subsections may be confusing.
In order to clarify this part, we will slightly reorganize the distribution of the different paragraphs and modified the title of the subsections as follows:
- 5.1 Paleoclimatic interpretation of $^3$He signals
- 5.2 Sources of uncertainty: this section contains now three subsections focusing on the methodological uncertainties associated with 1) the estimates of modern EDTs; 2) the interpretation of cosmogenic nuclide measurements; 3) the characterization of $^3$He diffusion kinetics
- 5.3 Potential role of permafrost processes